# Identification of 1,400 metabolites as mediators of obesity in 473 gut microbiota taxa: a mediation Mendelian randomization study

Xiaomin Li,[1] Qike Wu,[2] Shan-peng Liu[2]

**ABSTRACT** Obesity is a global health problem driven by genetic, endocrine, and environmental factors. Gut microbiota significantly influences obesity, yet causal relationships and underlying pathways remain elusive. The objective of the study was to investigate causal relationships between gut microbiota, metabolites, and obesity; elucidate potential pathways mediating obesity onset; identify novel genes; and explore the impact of plasma proteins on obesity risk. Bidirectional and two-sample Mendelian randomization (MR) were used to explore causal relationships. Mediation analyses identified mechanisms linking gut microbiota, metabolites, and obesity. Pathway analyses and protein-protein interaction assessed genetic and protein associations. The MR analysis results identified 11 gut microbiota species with causal associations with obesity, 69 metabolites that were significantly causally related to obesity, and seven bacteria with causal relationships, mediated by metabolites. Single nucleotide polymorphisms (SNP)-related gene set enrichment analysis revealed clustering in a concentration of genes enriched for phosphatidylinositol 3-kinase (PI3K)/protein kinase B (PKB/AKT) and plasma membrane-related signaling pathways. Fms-related receptor tyrosine kinase 1 (FLT1), growth-associated protein 43 (GAP43), and SLIT and NTRK-like family member 1 (SLITRK1) plasma proteins had protective effects against obesity. This study revealed causal links between gut bacteria, metabolites, and obesity, and identified potential therapeutic targets. Findings deepen understanding of obesity's complex mechanisms and suggest novel prevention and treatment strategies, emphasizing the gut microbiota and treatment targets.

**IMPORTANCE** This study pioneered the use of genetic approaches and mediator analyses to confirm the causal relationship between gut microbiota, metabolites, and obesity, and also explored how these factors work together to promote obesity through specific signaling pathways and protein interactions. This finding provides a theoretical basis and potential targets for precision medicine strategies against obesity, which is of great clinical significance. In addition, the identification of protective plasma proteins as biomarkers for obesity prevention opens up new avenues for tailoring obesity intervention strategies

**KEYWORDS** obesity, gut microbiota, metabolites, Mendelian randomization, genetic variation, mediation

Obesity, defined as an abnormal or excessive accumulation of fat that may impair health, is a chronic metabolic disease caused by a combination of genetic, endocrine, and environmental factors (1). From 1990 to 2022, the worldwide prevalence of obesity has increased substantially, with rates in adult males nearly tripling and in adult females more than doubling (2). Obesity is a major risk factor for a wide

Address correspondence to Shan-peng Liu, shanpengliu@mail.ccmu.edu.cn.

The authors declare no conflict of interest.

range of serious health problems, including metabolic diseases such as type 2 diabetes, cardiovascular disease (3), certain types of cancer (4), musculoskeletal disorders, and respiratory problems (5). This has become a major global public health concern, causing up to 5 million deaths per year attributable to overweight or obesity (6). The human gut harbors a diverse microbial community, collectively referred to as the gut microbiota, consisting of approximately 100 trillion microorganisms, often described as the "second genome" of the human body (7). This complex ecosystem plays a crucial role in all aspects of human physiology, including nutrient metabolism, immune system regulation, and maintenance of gut homeostasis (8–10).

Numerous studies have shown a strong association between gut microbiota and the development and progression of obesity (11). Study shows transplanting gut flora from normal diet and exercise mice into obese mice improves obesity symptoms and reduces body weight (12). A 2022 study established a link between the gut bacteria *Ruminococcus gnavus* and obesity in a large Norwegian cohort (13). The genus *Akkermansia* was found to have a protective effect against childhood obesity and lower body mass index (14). Metabolomics studies have revealed that obese patients frequently suffer from metabolic disorders, enhancing our understanding of obesity's pathogenesis (15, 16). Additionally, analyses of their metabolites and pathway characteristics provide valuable insights into the mechanisms underlying obesity.

Although previous studies have identified associations between gut microbiota, metabolome, and obesity, uncertainty remains regarding the specific causal relationships and their respective mediating roles (17, 18). Mendelian randomization (MR) uses genetic variants as instrumental variables (IVs) to assess causal links between exposures and outcomes, minimizing confounding factors (19). Furthermore, mediation analyses can be employed to ascertain the manner in which an exposure affects an outcome, namely through the mediation of variables (20). Consequently, we propose to conduct mediation analyses based on publicly available genome-wide association study (GWAS) pooled data. The objective is to assess potential causal relationships between the gut microbiota, plasma metabolites, and obesity and to identify specific pathways from the gut microbiota to plasma metabolites that mediate obesity onset. This may facilitate a more comprehensive understanding of the pathogenesis of obesity and offer novel insights into prevention and treatment strategies.

## MATERIALS AND METHODS

### Study design

In this study, we adopted a multi-faceted approach to unraveling the complex interplay between gut microbiota, metabolites, and obesity: first, we leveraged a bidirectional MR framework to investigate the potential causal relationships between gut microbiota, metabolites, and obesity. Next, we employed a two-sample MR methodology to delve deeper into the causal links between specific gut bacteria, metabolites, and obesity, focusing on the factors previously identified as causally associated with obesity. We then applied mediated MR techniques to elucidate the potential causal pathways involving obesity, the gut microbiota, and metabolites. Furthermore, we conducted a targeted search for novel genes associated with the identified aetiological genetic variants to further illuminate the underlying pathways and targets related to obesity. Intriguingly, we also discovered that three plasma proteins found to have a causal relationship with obesity exhibited a risk-reducing effect on obesity (Fig. 1).

### Data sources

In this study, the NHGRI-EBI GWAS Catalog database (https://www.ebi.ac.uk/gwas/) was employed to extract summary statistics on gut microbiota, with the accession numbers GCST90032172 to GCST90032644 serving as the search parameters. A cohort study was conducted involving 5,959 individuals from Finland. The data were determined through

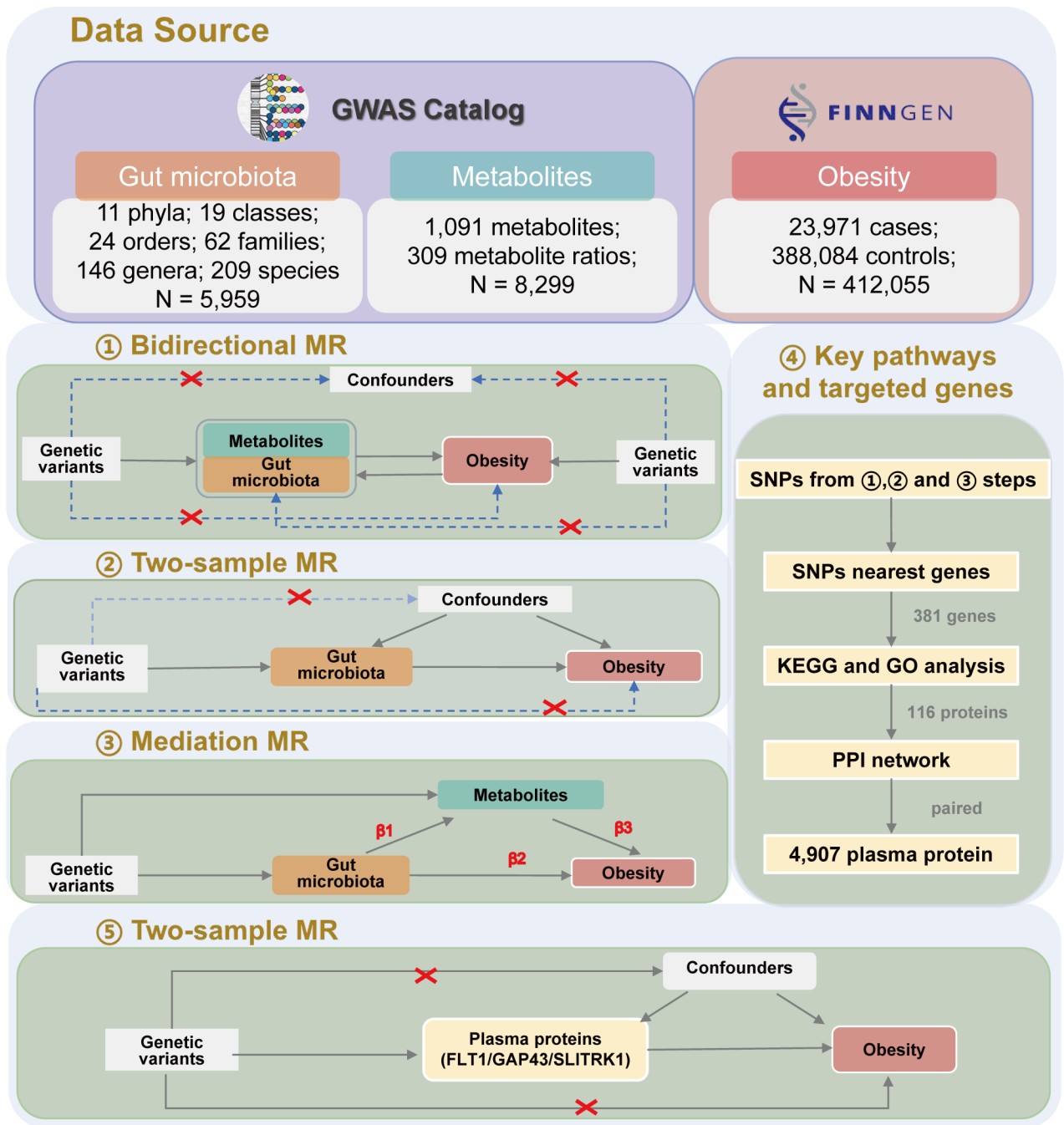

**FIG 1** The investigative process of this study is anchored in the utilization of MR analysis. Confounders encompass any identifiable or unidentifiable elements that might mislead the perceived link between exposure and outcome. SNP: single nucleotide polymorphisms; β: a beta value greater than zero implies a positive correlation between the SNP and the exposure, while a beta value less than zero denotes a negative correlation; KEGG: Kyoto Encyclopedia of Genes and Genomes; GO: Gene Ontology; PPI: protein-protein interaction.

shotgun metagenomic sequencing of fecal samples, encompassing a total of 473 taxonomic units. These included 11 phyla, 19 classes, 24 orders, 62 families, 146 genera, and 209 species (21). In addition, circulating metabolites were obtained for this study using the GWAS data sets on the human metabolome, including 1,091 blood metabolites and 309 metabolite ratios, from the analysis of 8,299 participants and approximately 15.4 million single nucleotide polymorphisms (SNPs) in the Canadian Longitudinal Study on Aging cohort. The full GWAS summary statistics for these 1,400 blood biomarkers are

publicly available in the GWAS Catalog (https://www.ebi.ac.uk/gwas/studies/GCST90199621-902010209) (22). The obesity outcome data is sourced from the R10 version of the comprehensive Finnish database (https://www.finngen.fi/) (23), these data include data from 412,055 participants (https://storage.googleapis.com/finngen-public-data-r10/summary_stats/finngen_R10_E4_OBESITY.gz). Plasma protein data were obtained from a comprehensive analysis of a large-scale protein quantitative trait loci (pQTL) study of 35,559 individuals in Iceland. The study focused on genetic relationships with the levels of 4,907 circulating proteins (24).

## Instrumental variable selection

The threshold set for SNPs associated with gut microbiota, metabolites, and plasma protein was $P < 1 \times 10^{-5}$, with linkage disequilibrium (LD) checks performed for these SNPs ($r^2 = 0.001$, kb = 10,000). In preparation for inverse MR validation, the threshold set for SNPs associated with obesity data was $P < 5 \times 10^{-8}$ ($r^2 = 0.01$, kb = 5,000) to obtain a sufficient number of SNPs for analysis. The $r^2$ or kb values indicate the degree of linkage imbalance between two loci, suggesting that the allele frequencies of these loci are not independent but somewhat correlated when LD is present. To avoid potential SNPs related to obesity-related diseases, we cross-checked the SNPs using the Phenoscan website (http://www.phenoscanner.medschl.cam.ac.uk/). Additionally, we utilized IEU Open GWAS (https://gwas.mrcieu.ac.uk/) and GWAS Catalog (https://www.ebi.ac.uk/gwas/) to identify all phenotypes associated with specific SNPs, excluding any SNPs related to confounding factors. To assess IV strength and reduce weak instrument bias, the f-statistic for each SNP was calculated, excluding those with F values < 10 (25).

The study used SNPs as IVs to ensure that the selected exposures met the three basic assumptions of MR analyses: (1) genetic variant must reliably associate with the risk factor; (2) genetic variant must not associate with any known or unknown confounders; (3) genetic variant must influence the outcome only through the risk factor and not through any direct causal pathway. Any IVs that contradicted these three assumptions were excluded from the analyses.

## MR analysis and sensitivity analysis

The study employed the Inverse Variance Weighted (IVW) approach (26) as the primary method for estimating the composite casual effect, with complementary calculations using methods such as MR-Egger, Weighted Mode, Weighted Median, and MR-PRESSO (27, 28), each of which made different assumptions about the potential for a multiplicity of effects. The complementary methods yielded results that were consistent with the IVW estimates, thereby enhancing the credibility of the findings. In this study, both MR-Egger and MR-Presso were used to assess the presence of horizontal pleiotropy ($P < 0.05$ was considered indicative). In addition, Cochran's Q test was used to assess heterogeneity among the selected SNPs ($P < 0.05$ indicated heterogeneity).

Mediation proportions were calculated using the formula (β1 × β3)/β2. In this formula, β2 represents the total effect derived from the first analysis, β1 represents the influence of gut microbiota on mediators (metabolites), and β3 represents the influence of mediators on obesity. The β value indicates the effect size of the SNP, with β > 0 indicating a positive correlation between the SNP and the exposure, whereas β < 0 indicates a negative correlation with obesity. Standard errors and CIs were calculated using delta methods. All analyses were performed with the statistical software R (version 4.3.3), using the TwoSampleMR (version 0.5.11) and MR-PRESSO (version 1.0) packages.

## Statistics and analysis of genes associated with obesity, gut microbiota, and metabolites

Following bidirectional MR and two-sample MR analyses, we identified gut microbiota/metabolism-related SNPs causally associated with obesity. We used the vautils software package (version 0.1.0) (https://github.com/oyhel/vautils) to identify the genes

| Bacterial taxa (Exposure) | Outcome | nSNP | pval | | OR(95%CI) | Q | Q_df | Q_pval | Egger_intercept | E_se | E_pval |
|---|---|---|---|---|---|---|---|---|---|---|---|
| f_Fibrobacteraceae | obesity | 16 | 0.025 | | 1.47 (1.05 - 2.05) | 14.712 | 15 | 0.472 | 0.007 | 0.008 | 0.357 |
| c_Thermococci | obesity | 20 | 0.019 | | 1.33 (1.05 - 1.69) | 23.389 | 19 | 0.221 | -0.006 | 0.010 | 0.535 |
| o_UNC496MF | obesity | 17 | 0.012 | | 1.33 (1.06 - 1.66) | 19.207 | 16 | 0.258 | 0.007 | 0.010 | 0.503 |
| g_Comamonas B | obesity | 18 | 0.015 | | 1.28 (1.05 - 1.56) | 8.385 | 17 | 0.958 | 0.004 | 0.008 | 0.656 |
| g_Bacillus U | obesity | 19 | 0.008 | | 1.25 (1.06 - 1.46) | 24.109 | 18 | 0.151 | -0.022 | 0.010 | 0.054 |
| f_GCA-900066135 sp900066135 | obesity | 19 | 0.015 | | 1.23 (1.04 - 1.45) | 24.159 | 18 | 0.150 | 0.008 | 0.009 | 0.367 |
| g_Geminocystis | obesity | 13 | 0.042 | | 0.84 (0.72 - 0.99) | 9.008 | 12 | 0.702 | 0.008 | 0.009 | 0.411 |
| o_UBA2922 sp900313925 | obesity | 16 | 0.011 | | 0.82 (0.71 - 0.96) | 10.203 | 15 | 0.807 | 0.006 | 0.010 | 0.588 |
| g_Dokdonella | obesity | 23 | 0.005 | | 0.78 (0.66 - 0.93) | 24.823 | 22 | 0.306 | -0.005 | 0.009 | 0.622 |
| o_Sporomusales | obesity | 14 | 0.004 | | 0.62 (0.45 - 0.86) | 14.377 | 13 | 0.348 | 0.002 | 0.009 | 0.863 |
| f_Acetobacteraceae | obesity | 14 | 0.011 | | 0.61 (0.42 - 0.90) | 7.947 | 13 | 0.847 | -0.008 | 0.010 | 0.441 |

p<0.05 was considered statistically significant

0 0.5 1 1.5 2

← protective factor risk factor →

**FIG 2** Assessing MR causality between gut microbiota and obesity. Exposure: 473 gut microbiota; outcome: obesity; the prefix "c_/o_/f_/g_" represents the following taxonomic categories, respectively: class, order, family, genus; nSNP: number of single nucleotide polymorphisms; method: inverse variance weighting; OR: odds ratio; CI: confidence interval, OR > 1 exposure as risk factor for outcome, OR < 1 exposure as protective factor for outcome. Q: heterogeneity analysis, Q_df: freedom of heterogeneity analysis. Q_pval < 0.05, indicating heterogeneity. Egger_intercept: pleiotropic analysis intercept; E_se: standard error of pleiotropic analysis; E_pval < 0.05 indicates the presence of pleiotropy.

most closely associated with the relevant SNPs. The find_nearest_gene function in the vautils package allows for the rapid identification of the nearest genes. If an SNP appeared to correspond to more than one gene, the closest gene was selected, duplicates were removed and all genes associated with the SNP were merged.

To explore the pathways and molecular mechanisms associated with SNPs-recently genes, we used the Kyoto Encyclopedia of the Genome (KEGG) pathway and Gene Ontology (GO) functional enrichment analyses, which are essential for understanding gene function and revealing genomic insights. In KEGG enrichment analysis, we obtained gene annotations from the KEGG API (https://www.kegg.jp/kegg/rest/keggapi.html) as background sets and analyzed them using the ClusterProfiler R package (version 4.12.0). GO enrichment analysis, on the other hand, was performed using the org.Hs.eg.db R package (version 3.19.1) to analyze SNPs for recent genes. These analyses enabled us to identify and evaluate enrichment pathways and molecular mechanisms associated with the most recent SNP-related genes in our study.

A protein-protein interaction (PPI) network was constructed using the STRING database (https://string-db.org/) and imported into Cytoscape software (version 3.10.0) for further analysis (29). The CytoHubba plugin, in combination with degree topological analysis methods, was utilized to identify key proteins within the PPI network (30).

## RESULTS

### Bidirectional MR analysis of gut microbiota for obesity

In our initial analysis, we investigated the causal relationship between 473 gut microbiota variables as exposure factors and obesity by IVW primary method (Fig. 2). A total of 11 gut microbiota (one class, three families and orders, four genera) were found to have a causal effect. The results indicate that the f_*Acetobacteraceae* (OR = 0.61, P = 0.011), g_*Dokdonella* (OR = 0.78, P = 0.005), g_*Geminocystis* (OR = 0.85, P = 0.042), o_*Sporomusales* (OR = 0.62, P = 0.004), and o_*UBA2922* sp900313925 (OR = 0.82, P = 0.011) were negatively associated with the risk of obesity, while c_*Thermococci* (OR = 1.33, P = 0.019), f_*Fibrobacteraceae* (OR = 1.46, P = 0.025), f_*GCA-900066135* sp900066135 (OR = 1.23, P = 0.015), g_*Bacillus U* (OR = 1.24, P = 0.008), g_*Comamonas B* (OR = 1.28, P = 0.015), O_*UNC496MF* (OR = 1.33, P = 0.012) were positively associated with the risk of obesity (all P < 0.05) (Fig. 2). In the sensitivity analyses, we assessed the potential issues of horizontal pleiotropy and heterogeneity. The test for horizontal pleiotropy was conducted using a combination of the MR-Egger and Inverse-Variance Weighted methods. The results did not indicate the horizontal pleiotropy in the above findings. Additionally, the Cochrane Q test showed no evidence of significant heterogeneity across all the reported results. The

| Exposure | Bacterial taxa (Outcome) | nSNP | pval | | OR(95%CI) | Q | Q_df | Q_pval | Egger_intercept | E_se | E_pval |
|---|---|---|---|---|---|---|---|---|---|---|---|
| obesity | f_GCA-900066135 sp900066135 | 299 | 0.040 | | 1.02 (1.00 - 1.03) | 278.491 | 298 | 0.785 | -0.001 | 0.001 | 0.465 |
| obesity | g_Geminocystis | 299 | 0.062 | | 1.02 (1.00 - 1.03) | 289.721 | 298 | 0.624 | 0.001 | 0.001 | 0.559 |
| obesity | g_Bacillus U | 299 | 0.168 | | 1.01 (0.99 - 1.03) | 321.931 | 298 | 0.163 | 0.000 | 0.001 | 0.797 |
| obesity | c_Thermococci | 299 | 0.356 | | 1.00 (0.99 - 1.01) | 313.080 | 298 | 0.263 | 0.000 | 0.001 | 0.718 |
| obesity | g_Comamonas B | 299 | 0.562 | | 1.00 (0.99 - 1.02) | 338.929 | 298 | 0.051 | 0.001 | 0.001 | 0.463 |
| obesity | f_Fibrobacteraceae | 299 | 0.786 | | 1.00 (0.99 - 1.01) | 326.829 | 298 | 0.121 | -0.001 | 0.001 | 0.089 |
| obesity | 0_Sporomusales | 299 | 0.698 | | 1.00 (0.99 - 1.01) | 304.268 | 298 | 0.389 | -0.001 | 0.001 | 0.241 |
| obesity | 0_Sporomusales | 299 | 0.698 | | 1.00 (0.99 - 1.01) | 304.268 | 298 | 0.389 | -0.001 | 0.001 | 0.241 |
| obesity | 0_Sporomusales | 299 | 0.698 | | 1.00 (0.99 - 1.01) | 304.268 | 298 | 0.389 | -0.001 | 0.001 | 0.241 |
| obesity | f_Acetobacteraceae | 299 | 0.020 | | 0.99 (0.98 - 1.00) | 338.654 | 298 | 0.052 | 0.000 | 0.001 | 0.729 |
| obesity | g_Dokdonella | 299 | 0.017 | | 0.98 (0.97 - 1.00) | 280.835 | 298 | 0.755 | -0.002 | 0.001 | 0.130 |

p<0.05 was considered statistically significant

0.96  0.98  1  1.02  1.04

protective factor  risk factor

FIG 3 Assessing reverse MR causality between gut microbiota and obesity. Exposure: obesity; outcome: 473 gut microbiota; the prefix "c_/o_/f_/g_" represents the following taxonomic categories, respectively: class, order, family, genus; nSNP: number of single nucleotide polymorphisms; method: inverse variance weighting; OR: odds ratio; CI: confidence interval, OR > 1 exposure as risk factor for outcome, OR < 1 exposure as protective factor for outcome. Q: heterogeneity analysis, Q_df: freedom of heterogeneity analysis. Q_pval < 0.05, indicating heterogeneity. Egger_intercept: pleiotropic analysis intercept; E_se: standard error of pleiotropic analysis; E_pval < 0.05 indicates the presence of pleiotropy.

additional four analysis methods, namely MR-Egger, Weighted Mode, Weighted Median, and MR-PRESSO can be found in Table S1.

To prevent reverse causality, reverse MR analyses were performed to explore potential causal effects between obesity and forward-significant bacteria. The results showed that f_*Acetobacteraceae* (OR = 0.99, $P$ = 0.017), f_*GCA-900066135 sp900066135* (OR = 1.02, $P$ < 0.040), and g_*Dokdonella* (OR = 0.98, $P$ = 0.017) were associated with obesity, with evidence of inverse causality observed. No inverse causality was found in any of the remaining eight species (all $P$ > 0.05) (Fig. 3). The results presented in the pictures were mainly generated by the IVW analysis method, and the content of the results of the remaining four analyses is presented in Table S2.

## Bidirectional MR analysis of metabolites for obesity

In order to gain further insight into the relationship between 1,400 metabolites as exposure variables and obesity, we employed the IVW approach to identify 60 known metabolites and nine unknown metabolites that were significantly associated with obesity. Among the 60 known metabolites, 33 were associated with an increased risk of obesity. The metabolites influenced by Glutamine conjugate of $C_6H_{10}O_2$ (OR = 1.12, $P$ = 0.006) and X-12104 (OR = 1.12, $P$ = 0.002) were the most significantly associated with obesity. The remaining 27 metabolites were associated with a reduced risk of obesity (all $P$ < 0.05). No horizontal pleiotropy was found in the above results, and the Cochrane Q test showed no heterogeneity in all results (all $P$ > 0.05) (Fig. 4).

To mitigate the risk of reverse causality, our research conducted reverse MR analyses. The results demonstrated reverse causality between 13 circulating metabolites and obesity (all $P$ < 0.05). In contrast, no reverse causality was observed among the remaining 57 metabolites (all $P$ > 0.05), and the $P$-values of both the heterogeneity and multiplicity analyses were greater than 0.05 (Fig. 5 ). The above revealed a significant causal relationship between metabolites and obesity using IVW approach, which was further refined by four additional analytical approaches (Tables S3 and S4).

## Causal effects of possible mediators on obesity

To further understand the connections between gut microbiota/metabolites and obesity, our analysis examined the links between pre-identified gut microbiota (Fig. 2) as potential exposures and the metabolites associated with obesity (Fig. 4) as potential outcomes. Using the two-sample MR method, we obtained the MR results for gut microbiota and metabolites connected with obesity by utilizing the IVW method (all $P$ < 0.05) (Fig. 6). The mediator proportion was an indicator used to evaluate the effect of

| levels | Metabolites (Exposure) | nSNP | pval | OR(95%CI) | Q | Q_df | Q_pval | Egger_intercept | E_se | E_pval |
|---|---|---|---|---|---|---|---|---|---|---|
| Amino Acid | Glutamine conjugate of C6H10O2 (1) | 15 | 0.006 | 1.12 (1.03 - 1.21) | 17.541 | 14 | 0.228 | 0.021 | 0.011 | 0.083 |
| | Glutamate | 21 | 0.008 | 1.10 (1.02 - 1.18) | 28.142 | 20 | 0.106 | -0.009 | 0.010 | 0.415 |
| | Glutamate to kynurenine ratio | 24 | 0.012 | 1.08 (1.02 - 1.15) | 32.692 | 23 | 0.087 | 0.010 | 0.009 | 0.264 |
| | Glucuronate | 15 | 0.024 | 1.08 (1.01 - 1.15) | 9.287 | 14 | 0.812 | 0.006 | 0.009 | 0.476 |
| | Cysteine to alanine ratio | 28 | 0.012 | 1.06 (1.01 - 1.12) | 29.453 | 27 | 0.339 | 0.004 | 0.007 | 0.543 |
| | Argininine | 27 | 0.010 | 1.06 (1.01 - 1.11) | 26.497 | 26 | 0.436 | -0.005 | 0.007 | 0.500 |
| | Glycosyl-N-palmitoyl-sphingosine (d18:1/16:0) | 22 | 0.029 | 1.05 (1.00 - 1.10) | 24.094 | 21 | 0.289 | -0.009 | 0.007 | 0.185 |
| | Alpha-ketoglutarate to kynurenine ratio | 26 | 0.034 | 1.05 (1.00 - 1.10) | 29.506 | 25 | 0.243 | 0.009 | 0.005 | 0.131 |
| | Histidine to glutamine ratio | 30 | 0.047 | 0.95 (0.91 - 1.00) | 30.146 | 29 | 0.407 | 0.002 | 0.005 | 0.777 |
| | Alpha-ketoglutarate to glutamate ratio | 32 | 0.016 | 0.95 (0.91 - 0.99) | 23.449 | 31 | 0.832 | -0.005 | 0.004 | 0.225 |
| | 2-hydroxyphenylacetate | 27 | 0.001 | 0.93 (0.90 - 0.97) | 21.869 | 26 | 0.696 | 0.000 | 0.007 | 0.971 |
| | Gamma-tocopherol/beta-tocopherol | 24 | 0.000 | 0.88 (0.83 - 0.94) | 25.919 | 23 | 0.305 | -0.010 | 0.007 | 0.176 |
| Cofactors and Vitamins | N2,n2-dimethylguanosine | 18 | 0.005 | 1.10 (1.03 - 1.17) | 16.612 | 17 | 0.481 | 0.007 | 0.009 | 0.454 |
| | 4-hydroxyphenylacetate | 22 | 0.016 | 1.06 (1.01 - 1.11) | 18.501 | 21 | 0.617 | -0.008 | 0.007 | 0.248 |
| | Caffeine to theophylline ratio | 28 | 0.040 | 0.95 (0.90 - 1.00) | 31.965 | 27 | 0.233 | -0.003 | 0.008 | 0.697 |
| Lipid | Arginine to citrulline ratio | 32 | 0.001 | 1.09 (1.04 - 1.16) | 41.832 | 31 | 0.093 | 0.000 | 0.008 | 0.984 |
| | Maltose in coronary artery disease | 21 | 0.002 | 1.09 (1.03 - 1.16) | 24.409 | 20 | 0.225 | -0.011 | 0.007 | 0.160 |
| | Adenosine 5'-diphosphate (ADP) to mannitol to sorbitol ratio | 20 | 0.001 | 1.08 (1.03 - 1.13) | 19.003 | 19 | 0.457 | -0.005 | 0.009 | 0.566 |
| | Sphingomyelin (d18:2/18:1) | 16 | 0.045 | 1.07 (1.00 - 1.14) | 9.618 | 15 | 0.843 | 0.004 | 0.014 | 0.773 |
| | Citrate to oxalate (ethanedioate) ratio | 16 | 0.047 | 1.07 (1.00 - 1.14) | 16.185 | 15 | 0.370 | -0.007 | 0.011 | 0.534 |
| | 4-acetaminophen sulfate | 25 | 0.001 | 1.07 (1.03 - 1.11) | 17.553 | 24 | 0.824 | 0.007 | 0.008 | 0.410 |
| | Isovalerylcarnitine (C5) | 24 | 0.014 | 1.06 (1.01 - 1.11) | 10.410 | 23 | 0.988 | 0.002 | 0.009 | 0.872 |
| | 4-methylhexanoylglutamine | 23 | 0.005 | 1.06 (1.02 - 1.10) | 23.602 | 22 | 0.368 | -0.008 | 0.006 | 0.206 |
| | Mannose | 25 | 0.018 | 1.06 (1.01 - 1.11) | 24.147 | 24 | 0.453 | 0.011 | 0.006 | 0.064 |
| | Glutarate (C5-DC) to caprylate (8:0) ratio | 25 | 0.043 | 1.06 (1.00 - 1.11) | 34.688 | 24 | 0.073 | 0.002 | 0.007 | 0.736 |
| | Adenosine 5'-monophosphate (AMP) to urate ratio | 27 | 0.024 | 1.06 (1.01 - 1.11) | 23.545 | 26 | 0.602 | -0.002 | 0.006 | 0.746 |
| | Arginine to ornithine ratio | 29 | 0.047 | 1.04 (1.00 - 1.08) | 25.544 | 28 | 0.598 | 0.006 | 0.005 | 0.322 |
| | Ascorbic acid 3-sulfate | 20 | 0.027 | 1.04 (1.00 - 1.08) | 14.067 | 19 | 0.780 | -0.004 | 0.004 | 0.319 |
| | Benzoate to linoleoyl-arachidonoyl-glycerol (18:2 to 20:4) [2] ratio | 26 | 0.045 | 1.04 (1.00 - 1.08) | 36.512 | 25 | 0.064 | 0.003 | 0.005 | 0.613 |
| | Mannonate | 22 | 0.039 | 0.98 (0.95 - 1.00) | 16.185 | 21 | 0.759 | -0.003 | 0.004 | 0.396 |
| | Dihydrocaffeate sulfate (2) | 29 | 0.039 | 0.96 (0.92 - 1.00) | 23.938 | 28 | 0.685 | -0.006 | 0.006 | 0.325 |
| | Dodecenedioate (C12:1-DC) | 21 | 0.027 | 0.95 (0.90 - 0.99) | 15.584 | 20 | 0.742 | 0.009 | 0.008 | 0.265 |
| | Lactosyl-N-palmitoyl-sphingosine (d18:1/16:0) | 24 | 0.019 | 0.94 (0.90 - 0.99) | 28.967 | 23 | 0.181 | -0.006 | 0.007 | 0.358 |
| | 1-(1-enyl-palmitoyl)-2-oleoyl-gpc (p-16:0/18:1) | 24 | 0.003 | 0.94 (0.90 - 0.98) | 19.994 | 23 | 0.642 | -0.007 | 0.005 | 0.189 |
| | Arachidonate (20:4n6) to caffeine ratio | 22 | 0.017 | 0.94 (0.90 - 0.99) | 23.862 | 21 | 0.300 | -0.005 | 0.007 | 0.471 |
| | 1-(1-enyl-palmitoyl)-GPC (p-16:0) | 26 | 0.003 | 0.92 (0.87 - 0.97) | 32.751 | 25 | 0.137 | -0.008 | 0.008 | 0.309 |
| Metabolite ratios | X-12104 | 13 | 0.002 | 1.12 (1.04 - 1.20) | 14.081 | 12 | 0.296 | 0.016 | 0.009 | 0.118 |
| | X-12701 | 15 | 0.004 | 1.08 (1.02 - 1.13) | 11.677 | 14 | 0.632 | 0.001 | 0.007 | 0.849 |
| | X-21742 | 20 | 0.022 | 1.06 (1.01 - 1.12) | 21.286 | 19 | 0.321 | 0.007 | 0.007 | 0.343 |
| | X-24728 | 32 | 0.040 | 1.05 (1.00 - 1.10) | 34.968 | 31 | 0.285 | 0.001 | 0.007 | 0.825 |
| | Suberate (C8-DC) | 27 | 0.029 | 0.96 (0.93 - 1.00) | 20.161 | 26 | 0.784 | -0.004 | 0.005 | 0.519 |
| | Urate | 29 | 0.025 | 0.94 (0.90 - 0.99) | 38.539 | 28 | 0.089 | -0.005 | 0.007 | 0.484 |
| | X-11372 | 25 | 0.016 | 0.94 (0.90 - 0.99) | 24.662 | 24 | 0.424 | 0.002 | 0.006 | 0.766 |
| | X-25422 | 24 | 0.008 | 0.94 (0.90 - 0.98) | 22.175 | 23 | 0.510 | 0.010 | 0.007 | 0.136 |
| | X-12411 | 16 | 0.042 | 0.94 (0.88 - 1.00) | 25.676 | 15 | 0.042 | 0.003 | 0.007 | 0.729 |
| | X-23644 | 15 | 0.034 | 0.94 (0.88 - 0.99) | 14.072 | 14 | 0.444 | -0.003 | 0.007 | 0.734 |
| | Tyrosine | 29 | 0.005 | 0.94 (0.90 - 0.98) | 32.225 | 28 | 0.265 | 0.002 | 0.006 | 0.792 |
| | Taurine to cysteine ratio | 20 | 0.030 | 0.94 (0.88 - 0.99) | 16.387 | 19 | 0.631 | 0.003 | 0.008 | 0.690 |
| | X-24307 | 16 | 0.037 | 0.93 (0.87 - 1.00) | 17.221 | 15 | 0.306 | -0.016 | 0.009 | 0.084 |
| | Succinate to trans-4-hydroxyproline ratio | 16 | 0.005 | 0.93 (0.88 - 0.98) | 17.888 | 15 | 0.269 | -0.004 | 0.006 | 0.512 |
| | X-17351 | 17 | 0.023 | 0.92 (0.86 - 0.99) | 24.145 | 16 | 0.086 | -0.013 | 0.013 | 0.353 |
| | X-25109 | 29 | 0.000 | 0.91 (0.87 - 0.95) | 25.406 | 28 | 0.606 | 0.002 | 0.007 | 0.727 |
| Nucleotide | 2-methylserine | 26 | 0.002 | 1.03 (1.01 - 1.05) | 23.905 | 25 | 0.525 | -0.001 | 0.004 | 0.842 |
| | 5-hydroxyindole sulfate | 18 | 0.044 | 0.94 (0.88 - 1.00) | 22.649 | 17 | 0.161 | -0.001 | 0.009 | 0.880 |
| Unknown | N-alpha-acetylornithine | 21 | 0.012 | 1.07 (1.02 - 1.13) | 32.471 | 20 | 0.039 | 0.019 | 0.010 | 0.066 |
| | Pregnenediol sulfate (C21H34O5S) | 30 | 0.017 | 1.07 (1.01 - 1.12) | 35.423 | 29 | 0.191 | 0.009 | 0.006 | 0.131 |
| | N-palmitoyl-sphinganine (d18:0/16:0) | 23 | 0.037 | 1.06 (1.00 - 1.13) | 31.414 | 22 | 0.088 | 0.001 | 0.008 | 0.903 |
| | Serine to alpha-tocopherol ratio | 28 | 0.017 | 1.06 (1.01 - 1.12) | 35.194 | 27 | 0.134 | 0.005 | 0.007 | 0.429 |
| | Oleoyl-linoleoyl-glycerol (18:1/18:2) [2] | 28 | 0.032 | 1.05 (1.00 - 1.09) | 32.164 | 27 | 0.226 | -0.007 | 0.006 | 0.303 |
| | Paraxanthine to 5-acetylamino-6-formylamino-3-methyluracil ratio | 25 | 0.034 | 0.97 (0.95 - 1.00) | 24.454 | 24 | 0.436 | -0.008 | 0.005 | 0.123 |
| | N-oleoyltaurine | 21 | 0.025 | 0.95 (0.91 - 0.99) | 22.556 | 20 | 0.310 | -0.008 | 0.008 | 0.314 |
| | Phosphate to cysteine ratio | 24 | 0.034 | 0.94 (0.89 - 0.99) | 29.828 | 23 | 0.154 | -0.005 | 0.007 | 0.470 |
| | N-acetylmethionine | 16 | 0.038 | 0.94 (0.89 - 1.00) | 17.547 | 15 | 0.287 | 0.002 | 0.007 | 0.759 |
| Xenobiotics | Sphingomyelin (d18:1/24:1, d18:2/24:0) | 19 | 0.015 | 1.08 (1.02 - 1.16) | 15.027 | 18 | 0.660 | -0.016 | 0.009 | 0.103 |
| | (2,4 or 2,5)-dimethylphenol sulfate | 21 | 0.046 | 1.05 (1.00 - 1.11) | 23.908 | 20 | 0.246 | -0.002 | 0.010 | 0.831 |
| | Beta-hydroxyisovaleroylcarnitine | 36 | 0.044 | 1.04 (1.00 - 1.08) | 39.206 | 35 | 0.287 | 0.008 | 0.006 | 0.184 |
| | Metabolonic lactone sulfate | 31 | 0.015 | 1.03 (1.01 - 1.06) | 36.415 | 30 | 0.195 | 0.002 | 0.004 | 0.679 |
| | Cysteine | 15 | 0.007 | 0.94 (0.90 - 0.98) | 13.629 | 14 | 0.478 | 0.003 | 0.006 | 0.624 |
| | 4-methyl-2-oxopentanoate to 3-methyl-2-oxobutyrate ratio | 21 | 0.003 | 0.90 (0.85 - 0.97) | 20.068 | 20 | 0.454 | -0.003 | 0.009 | 0.764 |

p<0.05 was considered statistically significant

0.8　0.9　1　1.1　1.2

protective factor ← | → risk factor

**FIG 4** Evaluating MR causation between metabolites and obesity. Exposure: 1,400 metabolites; outcome: obesity; nSNP: number of single nucleotide polymorphisms; method: inverse variance weighting; OR: odds ratio; CI: confidence interval, OR > 1 exposure as risk factor for outcome, OR < 1 exposure as protective factor for outcome. Q: heterogeneity analysis, Q_df: freedom of heterogeneity analysis. Q_pval < 0.05, indicating heterogeneity. Egger_intercept: pleiotropic analysis intercept; E_se: standard error of pleiotropic analysis; E_pval < 0.05 indicates the presence of pleiotropy.

these mediators. Metabolites were screened to become reliable mediators of metabolite effects on obesity.

Based on the positive two-sample MR analysis results, we selected gut microbiota as the exposures and metabolites as the mediators to further mediate MR analyses and explore their effects on obesity. At the overall level, the effect of metabolites as mediators on the MR causality between gut microbiota and obesity was significant

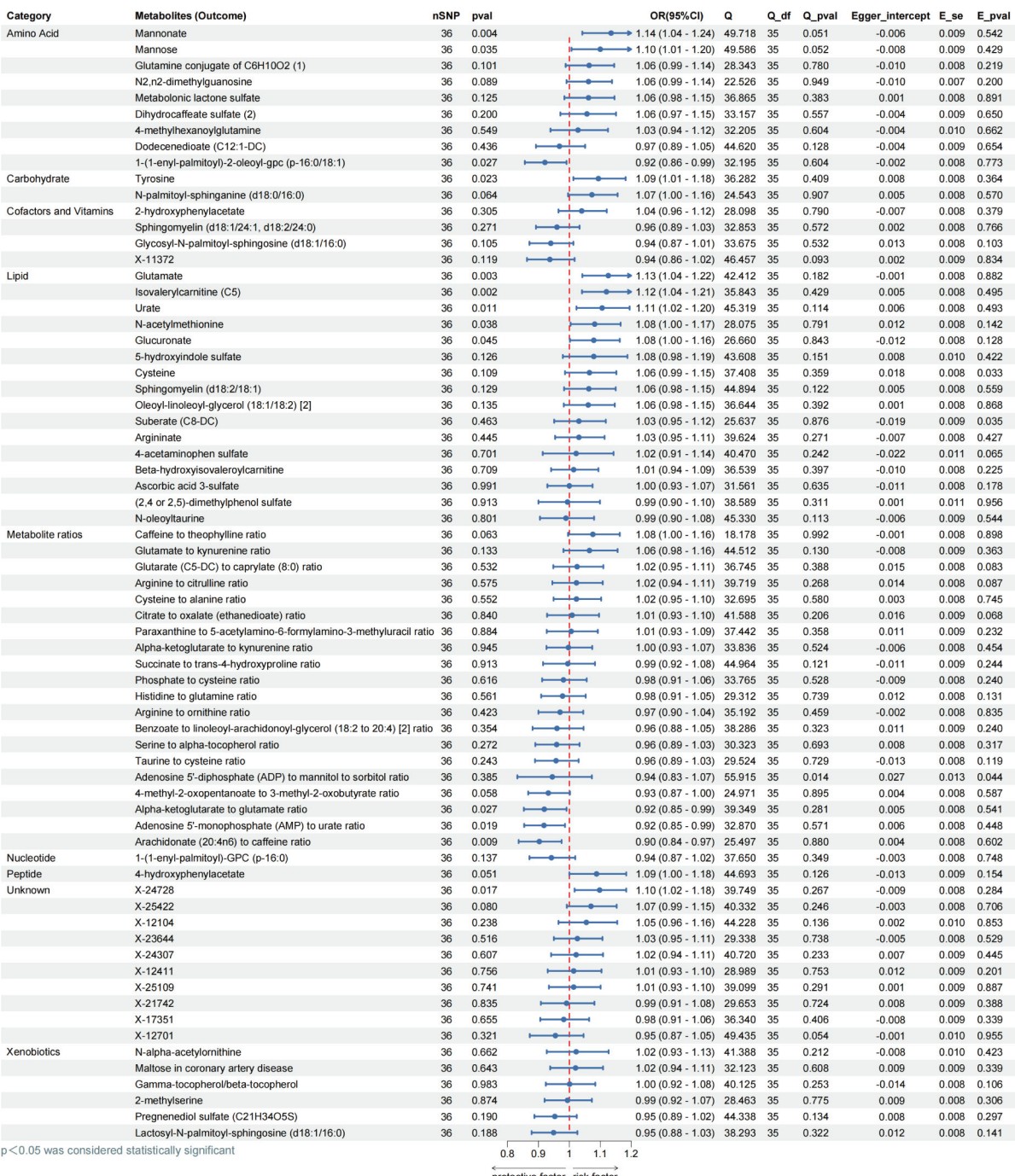

FIG 5 Evaluating reverse MR causation between metabolites and obesity. Exposure: obesity; Outcome: 1400 metabolites; nSNP: number of single nucleotide polymorphisms; method: inverse variance weighting；OR: odds ratio; CI: confidence interval, OR > 1 Exposure as risk factor for outcome, OR < 1 Exposure as protective factor for outcome. Q: Heterogeneity analysis，Q_df: Freedom of heterogeneity analysis. Q_pval < 0.05, indicating heterogeneity. Egger_intercept: Pleiotropic analysis intercept; E_se: Standard error of pleiotropic analysis; E_pval < 0.05 indicates the presence of pleiotropy.

(mediator proportion > 6%) (Table 1). Among the mediator proportions greater than 15%, "Maltose in coronary artery disease" mediated the causal relationship between the family *Fibrobacteraceae* and obesity (mediator proportion = 16.40%). Additionally, "X-12701" and "Glutamine conjugate of $C_6H_{10}O_2$ (1)" mediated the causal association between the operational taxonomic unit o_*UBA2922* sp900313925 and obesity, with mediation proportion of 18.10% and 25.10%, respectively (Table 1). The findings

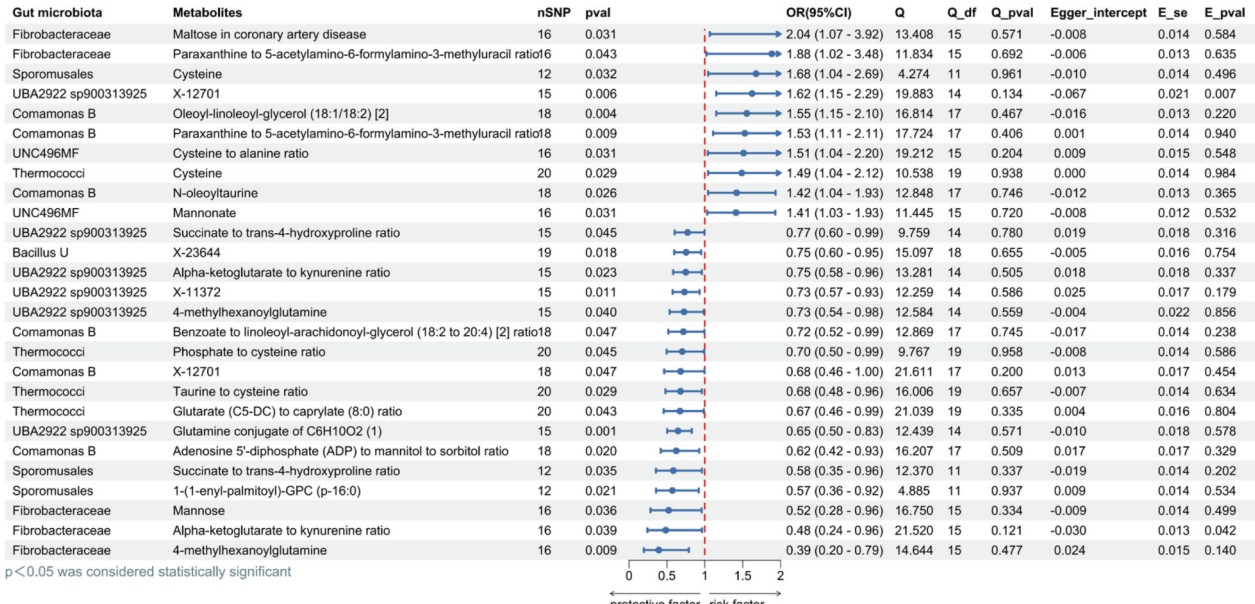

| Gut microbiota | Metabolites | nSNP | pval | | OR(95%CI) | Q | Q_df | Q_pval | Egger_intercept | E_se | E_pval |
|---|---|---|---|---|---|---|---|---|---|---|---|
| Fibrobacteraceae | Maltose in coronary artery disease | 16 | 0.031 | | 2.04 (1.07 - 3.92) | 13.408 | 15 | 0.571 | -0.008 | 0.014 | 0.584 |
| Fibrobacteraceae | Paraxanthine to 5-acetylamino-6-formylamino-3-methyluracil ratio | 16 | 0.043 | | 1.88 (1.02 - 3.48) | 11.834 | 15 | 0.692 | -0.006 | 0.013 | 0.635 |
| Sporomusales | Cysteine | 12 | 0.032 | | 1.68 (1.04 - 2.69) | 4.274 | 11 | 0.961 | 0.014 | 0.496 |
| UBA2922 sp900313925 | X-12701 | 15 | 0.006 | | 1.62 (1.15 - 2.29) | 19.883 | 14 | 0.134 | -0.067 | 0.021 | 0.007 |
| Comamonas B | Oleoyl-linoleoyl-glycerol (18:1/18:2) [2] | 18 | 0.004 | | 1.55 (1.15 - 2.10) | 16.814 | 17 | 0.467 | -0.016 | 0.013 | 0.220 |
| Comamonas B | Paraxanthine to 5-acetylamino-6-formylamino-3-methyluracil ratio | 18 | 0.009 | | 1.53 (1.11 - 2.11) | 17.724 | 17 | 0.406 | 0.001 | 0.014 | 0.940 |
| UNC496MF | Cysteine to alanine ratio | 16 | 0.031 | | 1.51 (1.04 - 2.20) | 19.212 | 15 | 0.204 | 0.009 | 0.015 | 0.548 |
| Thermococci | Cysteine | 20 | 0.029 | | 1.49 (1.04 - 2.12) | 10.538 | 19 | 0.938 | 0.000 | 0.014 | 0.984 |
| Comamonas B | N-oleoyltaurine | 18 | 0.026 | | 1.42 (1.04 - 1.93) | 12.848 | 17 | 0.746 | -0.012 | 0.013 | 0.365 |
| UNC496MF | Mannonate | 16 | 0.031 | | 1.41 (1.03 - 1.93) | 11.445 | 15 | 0.720 | -0.008 | 0.012 | 0.532 |
| UBA2922 sp900313925 | Succinate to trans-4-hydroxyproline ratio | 15 | 0.045 | | 0.77 (0.60 - 0.99) | 9.759 | 14 | 0.780 | 0.019 | 0.018 | 0.316 |
| Bacillus U | X-23644 | 19 | 0.018 | | 0.75 (0.60 - 0.95) | 15.097 | 18 | 0.655 | -0.005 | 0.016 | 0.754 |
| UBA2922 sp900313925 | Alpha-ketoglutarate to kynurenine ratio | 15 | 0.023 | | 0.75 (0.58 - 0.96) | 13.281 | 14 | 0.505 | 0.018 | 0.018 | 0.337 |
| UBA2922 sp900313925 | X-11372 | 15 | 0.011 | | 0.73 (0.57 - 0.93) | 12.259 | 14 | 0.586 | 0.025 | 0.017 | 0.179 |
| UBA2922 sp900313925 | 4-methylhexanoylglutamine | 15 | 0.040 | | 0.73 (0.54 - 0.98) | 12.584 | 14 | 0.559 | -0.004 | 0.022 | 0.856 |
| Comamonas B | Benzoate to linoleoyl-arachidonoyl-glycerol (18:2 to 20:4) [2] ratio | 18 | 0.047 | | 0.72 (0.52 - 0.99) | 12.869 | 17 | 0.745 | -0.017 | 0.014 | 0.238 |
| Thermococci | Phosphate to cysteine ratio | 20 | 0.045 | | 0.70 (0.50 - 0.99) | 9.767 | 19 | 0.958 | -0.008 | 0.014 | 0.586 |
| Comamonas B | X-12701 | 18 | 0.047 | | 0.68 (0.46 - 1.00) | 21.611 | 17 | 0.200 | 0.013 | 0.017 | 0.454 |
| Thermococci | Taurine to cysteine ratio | 20 | 0.029 | | 0.68 (0.48 - 0.96) | 16.006 | 19 | 0.657 | -0.007 | 0.014 | 0.634 |
| Thermococci | Glutarate (C5-DC) to caprylate (8:0) ratio | 20 | 0.043 | | 0.67 (0.46 - 0.99) | 21.039 | 19 | 0.335 | 0.004 | 0.016 | 0.804 |
| UBA2922 sp900313925 | Glutamine conjugate of C6H10O2 (1) | 15 | 0.001 | | 0.65 (0.50 - 0.83) | 12.439 | 14 | 0.571 | -0.010 | 0.018 | 0.578 |
| Comamonas B | Adenosine 5'-diphosphate (ADP) to mannitol to sorbitol ratio | 18 | 0.020 | | 0.62 (0.42 - 0.93) | 16.207 | 17 | 0.509 | 0.017 | 0.017 | 0.329 |
| Sporomusales | Succinate to trans-4-hydroxyproline ratio | 12 | 0.035 | | 0.58 (0.35 - 0.96) | 12.370 | 11 | 0.337 | -0.019 | 0.014 | 0.202 |
| Sporomusales | 1-(1-enyl-palmitoyl)-GPC (p-16:0) | 12 | 0.021 | | 0.57 (0.36 - 0.92) | 4.885 | 11 | 0.937 | 0.009 | 0.014 | 0.534 |
| Fibrobacteraceae | Mannose | 16 | 0.036 | | 0.52 (0.28 - 0.96) | 16.750 | 15 | 0.334 | -0.009 | 0.014 | 0.499 |
| Fibrobacteraceae | Alpha-ketoglutarate to kynurenine ratio | 16 | 0.039 | | 0.48 (0.24 - 0.96) | 21.520 | 15 | 0.121 | -0.030 | 0.013 | 0.042 |
| Fibrobacteraceae | 4-methylhexanoylglutamine | 16 | 0.009 | | 0.39 (0.20 - 0.79) | 14.644 | 15 | 0.477 | 0.024 | 0.015 | 0.140 |

p<0.05 was considered statistically significant

0   0.5   1   1.5   2

protective factor   risk factor

**FIG 6** Evaluating MR causation between gut microbiota and metabolites based on data demonstrably associated with obesity. Exposure: obesity-related gut microbiota; outcome: obesity-related metabolites; nSNP: number of single nucleotide polymorphisms; method: inverse variance weighting; OR: odds ratio; CI: confidence interval, OR > 1 exposure as risk factor for outcome, OR < 1 exposure as protective factor for outcome. Q: heterogeneity analysis, Q_df: freedom of heterogeneity analysis. Q_pval < 0.05, indicating heterogeneity. Egger_intercept: pleiotropic analysis intercept; E_se: standard error of pleiotropic analysis; E_pval < 0.05 indicates the presence of pleiotropy.

highlight the complex relationships between the gut microbiota, metabolic factors, and susceptibility to obesity.

## Identification of relevant pathways and targets of gut microbiota and metabolites as causal factors associated with obesity

Based on the bidirectional MR and two-sample MR of the above procedure, we eliminated duplicate SNPs and identified 548 gut microbiota/metabolites-related SNPs causally associated with obesity, and 381 genes were obtained using the vautils package (Table S5).

KEGG pathway enrichment analysis of the genes associated with the obesity-related gut microbiota/metabolism SNPs revealed that the top enriched pathways were involved in signaling cascades, including the PI3K-Akt signaling pathway, axon guidance, phosphatidylinositol signaling system, and platelet activation. In addition, pathways related to metabolic processes, such as bile secretion and inositol phosphate metabolism, were also enriched (Fig. 7A). GO enrichment analysis further revealed that these genes were predominantly localized to plasma membrane-associated components, including the plasma membrane region, intrinsic and integral membrane proteins, and synaptic structures (Fig. 7B). These findings suggest that the gut microbiota-related genetic factors associated with obesity may exert their effects by modulating signaling cascades and membrane-associated processes, potentially affecting metabolic homeostasis and neuronal function.

The Degree algorithm for PPI analysis with Degree showed that 116 proteins interacted with 149 lines at a confidence interval threshold of 0.5 (Fig. 7C). In conclusion, this comprehensive analysis provides insight into the causal mechanisms underlying the relationship between gut microbiota, metabolites, and obesity, highlighting key pathways and potential molecular targets for further research and therapeutic intervention.

**TABLE 1** Metabolites as mediators of MR causation effects in gut microbiota and obesity

| Obesity (outcome) | Gut microbiota (exposure) | Metabolites (mediators) | Total effect | Direct effect A | Direct effect B | Indirect effect | Mediated proportion (%) |
|---|---|---|---|---|---|---|---|
| Obesity | g_Bacillus | X-23644 | 0.219 | −0.284 | −0.063 | 0.018 | 8.20 |
| | Comamonas B | Oleoyl-linoleoyl-glycerol (18:1/18:2) [2] | 0.248 | 0.441 | 0.047 | 0.021 | 8.40 |
| | f_Fibrobacteraceae | Maltose in coronary artery disease | 0.384 | 0.715 | 0.088 | 0.063 | 16.40 |
| | Sporomusales | Cysteine | −0.478 | 0.517 | −0.081 | −0.042 | 8.70 |
| | Thermococci | Phosphate to cysteine ratio | 0.287 | −0.354 | −0.067 | 0.024 | 8.30 |
| | | Glutarate (C5-DC) to caprylate (8:0) ratio | 0.287 | −0.397 | −0.058 | 0.023 | 8.00 |
| | o_UBA2922 sp900313925 | Glutamine conjugate of $C_6H_{10}O_2$ (1) | −0.195 | −0.437 | 0.112 | −0.049 | 25.10 |
| | | X-11372 | −0.195 | −0.312 | 0.074 | −0.023 | 11.90 |
| | | X-12701 | −0.195 | 0.485 | −0.073 | −0.035 | 18.10 |
| | | Alpha-ketoglutarate to kynurenine ratio | −0.195 | −0.291 | 0.057 | −0.017 | 8.50 |
| | | Succinate to trans-4-hydroxy-proline ratio | −0.195 | −0.257 | 0.05 | −0.013 | 6.50 |
| | o_UNC496MF | Cysteine to alanine ratio | 0.285 | 0.413 | 0.063 | 0.026 | 9.10 |

## Two-sample MR analysis of plasma proteins and obesity

Based on the results of the PPI analysis, 116 proteins were matched with 4,907 plasma proteins and successful pairing was achieved with 39 plasma proteins that had GWAS data. The study then conducted a two-sample MR analysis using these 39 plasma proteins to investigate their causal relationship with obesity.

The findings of the study revealed that three plasma proteins (FLT1, GAP43, and SLITRK1) were significantly associated with the risk of obesity and showed a causal relationship. FLT1, GAP43, and SLITRK1 remained significant under the MR hypothesis and were identified as having a cause-and-effect relationship with obesity according to the IVW method. FLT1 exhibited a causal relationship with obesity (OR: 0.939, 95% CI: 0.901–0.978, $P$ = 0.002), GAP43 showed a causal relationship with obesity (OR: 0.897, 95% CI: 0.806–0.998, $P$ = 0.046), and SLITRK1 was causally linked to obesity (OR: 0.855, 95% CI: 0.736–0.994, $P$ = 0.041). The findings suggest that these proteins reduce the risk of developing obesity.

The scatter plot provided an assessment of the causal effects of the relationship between plasma proteins and the risk of obesity. Forest maps were used to illustrate the causal effects of SNPs associated with individual plasma proteins on obesity, as observed in the MR_SingleSNP test. The leave-one-out test indicated that there was no bias in the estimate for SNPs with large effect sizes. Funnel diagrams, along with global tests of the IVW and MR Egger methods, ruled out the possibility of heterogeneity (Q_pval > 0.05) (Fig. 8A through C).

In summary, this study utilized MR analysis to investigate the causal relationship between plasma proteins and obesity. It identified three plasma proteins (FLT1, GAP43, and SLITRK1) that were significantly associated with obesity and showed a causal relationship. The findings contribute to our understanding of the role of these proteins in the development of obesity.

## DISCUSSION

Obesity is a complex, multifactorial condition driven by genetic, endocrine, and environmental factors. Increasing evidence suggests the gut microbiota and its associated metabolites play a crucial role in the development and progression of obesity (1, 11, 31). Unraveling these intricate causal relationships is crucial for addressing the

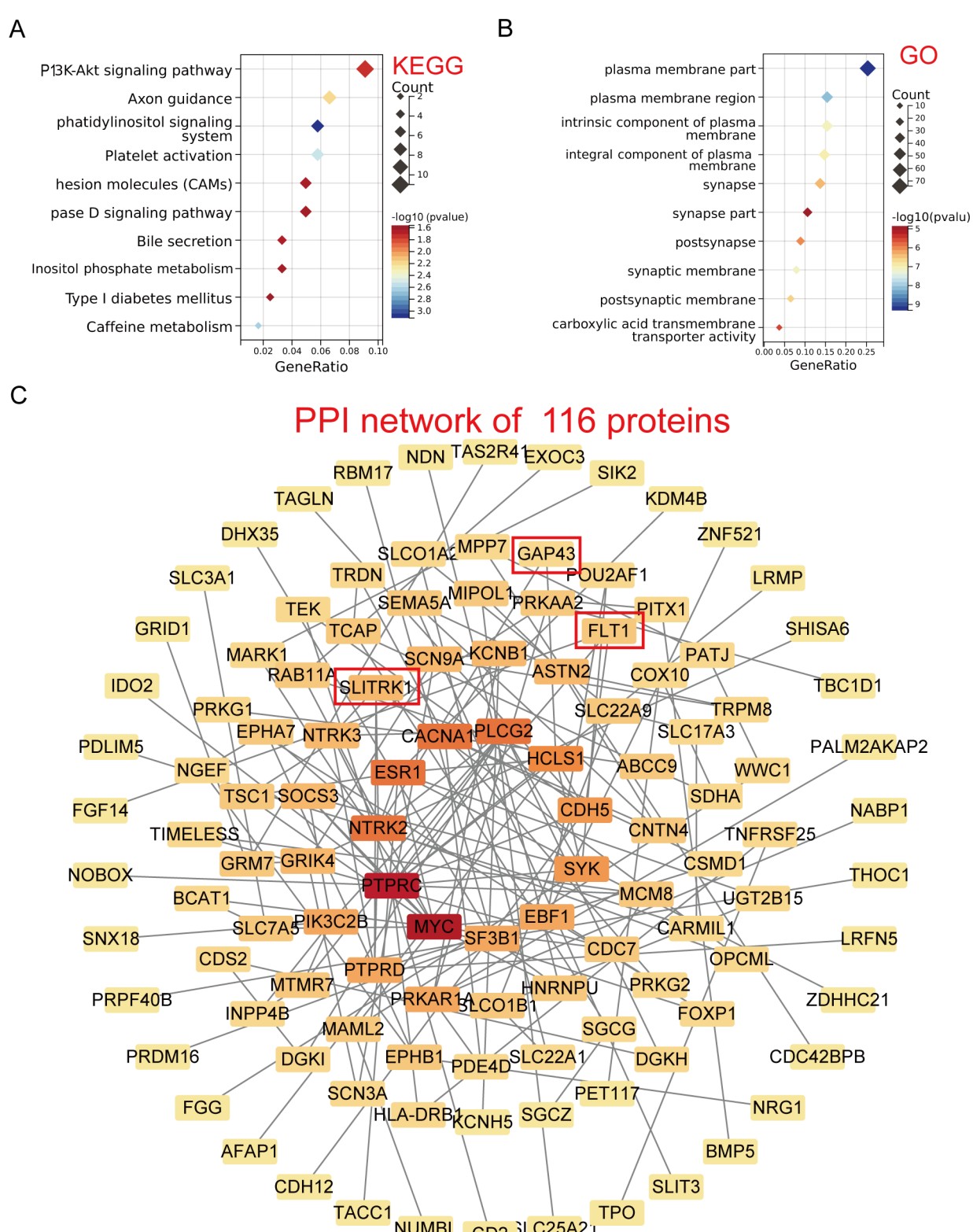

**FIG 7** Performing KEGG pathway and GO enrichment analysis on SNP-associated genes to investigate the causal relationship between gut microbiota, metabolites, and obesity. (A) KEGG. (B) GO. (C) Protein interactions were identified by the STRING database. The network is comprised of 149 edges and 116 nodes (PPI enrichment $P$ value = 1.0e$^{-16}$). Plasma proteins in red boxes have MR causality with obesity.

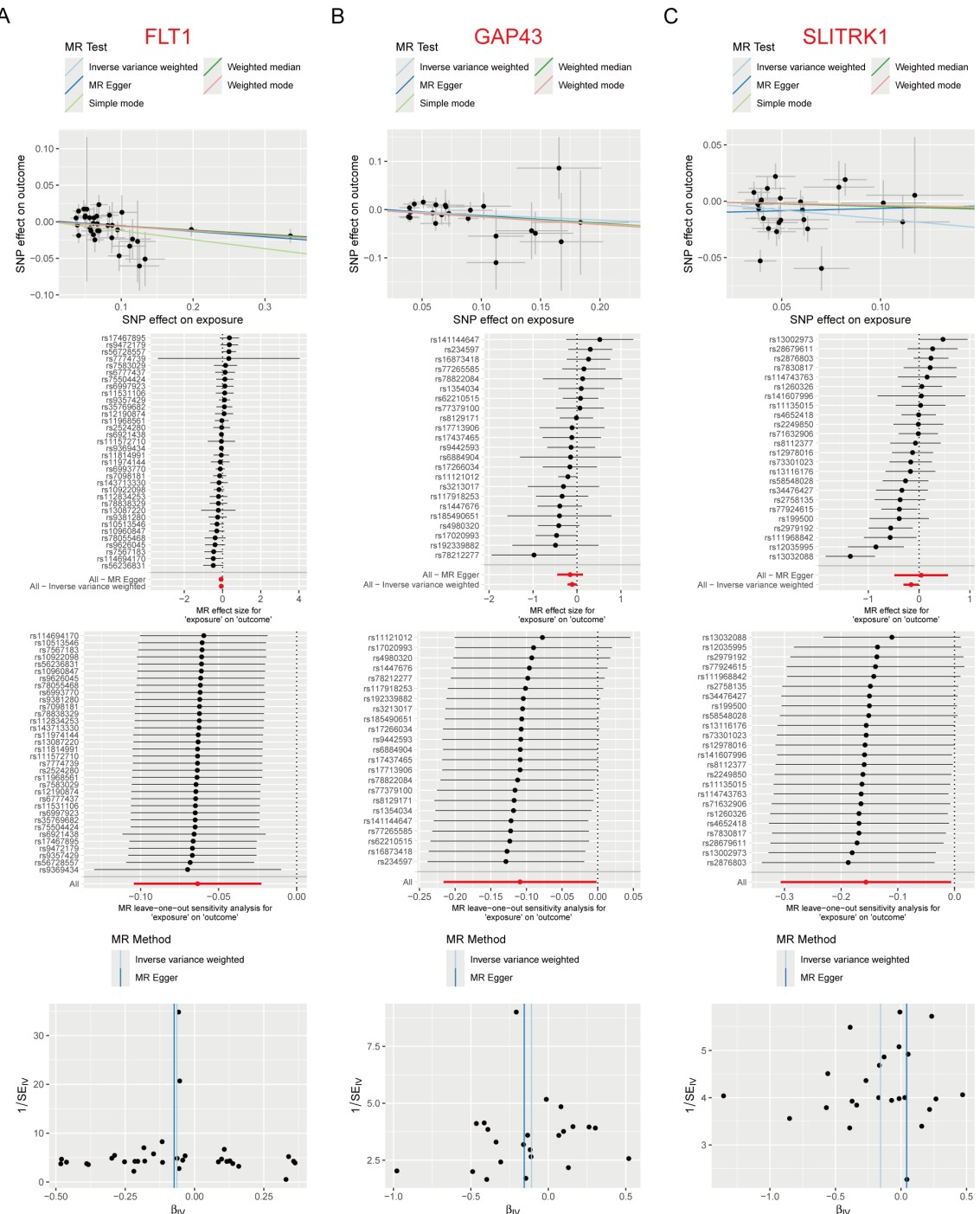

**FIG 8** MR analysis of plasma proteins and risk of obesity. (A) FLT1; (B) GAP43; (C) SLITRK1. The scatter plot shows the size (β) of the SNP effect on the outcome (Y-axis) and exposure (X-axis), along with the 95% confidence interval. Each dot represents a SNP that is used as a genetic instrument. The slope of the scatter plot represents the estimate from each of the five different MR methods used in the analysis. The SNP effect of each plasma protein associated with obesity is expressed as an SD change per 1 unit increase in the exposure. For the outcome (obesity), the effects are expressed as a logarithmic change in probability per 1 unit increase in the exposure. The forest plot presents the association between plasma proteins and obesity, displaying the estimated effect sizes and 95% confidence intervals. The "leave-one-out" analysis plot examines the robustness of the plasma protein-obesity associations by iteratively removing one SNP at a time and re-estimating the effects. A funnel plot visualizes the distribution of the associations between plasma proteins and obesity, which can be used to assess potential publication bias or small-study effects. MR: Mendelian randomization; SNP: single nucleotide polymorphisms; SD: standard deviation.

global obesity epidemic. We employed mediation MR analysis to investigate the strong association between the latest GWAS data on 473 gut microbiota and 1,400 plasma metabolites and obesity. We found evidence suggesting a causal relationship between seven bacterial species and obesity, with this relationship being mediated by metabolites.

The study utilized MR analysis to explore the causal relationships between gut microbiota, metabolites, plasma proteins, and obesity. The analysis identified specific gut microbiota, such as f_*Acetobacteraceae*, g_*Dokdonella*, g_*Geminocystis*, o_*Sporomusales*, and o_*UBA2922* sp900313925, which were negatively associated with obesity risk, while others like c_*Thermococci*, f_*Fibrobacteraceae*, and several bacteria were positively associated with obesity risk. These findings support the causal relationship between gut microbiota and obesity, based on the latest reliable data compared to past studies (32, 33). Several studies demonstrate a strong relationship between metabolites and obesity (34, 35). In the bidirectional MR analysis, 60 known metabolites were found to significantly increase obesity risk, with Glutamine conjugate of $C_6H_{10}O_2$ and X-12104 being the most strongly linked. Conversely, 27 metabolites showed a reduced obesity risk.

MR analyses reveal that metabolites such as oleoyl-linoleoyl-glycerol, phosphate to cysteine ratio, and alpha-ketoglutarate to kynurenine ratio mediate the relationship between gut microbiota and obesity. Gut bacteria can influence lipid metabolism and energy expenditure by producing molecules that affect fat storage (36). Additionally, adipocyte glutamine turnover is crucial for metabolic health, and the glutamine conjugate of $C_6H_{10}O_2$ may impact metabolic signaling pathways (37). Cysteine is emerging as a potential biomarker for obesity due to its role in antioxidant defense (38). Moreover, alpha-ketoglutarate supplementation promotes beige adipogenesis and reduces high-fat diet-induced obesity in middle-aged mice (39). Metabolites also interact with the gut-brain axis, affecting hunger, and energy regulation (40).

The KEGG and GO analyses demonstrate that genes associated with obesity-related gut microbiota and metabolism SNPs are primarily enriched in signaling pathways such as the phosphatidylinositol 3-kinase (PI3K) and protein kinase B (AKT) pathway, as well as in pathways related to metabolic processes like bile secretion and inositol phosphate metabolism. These genes are predominantly localized in membrane-associated regions, including the plasma membrane, integral membrane proteins, and synaptic structures. The PI3K-Akt signaling pathway plays a vital role in metabolic balance (41, 42). Thus, the results suggest that gut microbiota have an important impact on obesity by regulating metabolite mechanisms through the PI3K-Akt signaling pathway.

The Degree algorithm for PPI analysis further highlighted potential molecular targets for obesity. The study investigated the causal relationship between three plasma proteins (FLT1, GAP43, and SLITRK1) and obesity, finding that they have a protective effect against obesity risk. The study found that FLT1 promotes the proliferation of endothelial cells and maintains the internal environment of adipose tissue (43). Other studies examined the effects of high-fat diet-induced obesity on peripheral nerve regeneration and the levels of GAP43, an intrinsic determinant of neuronal development and plasticity, in rats (44, 45). Interestingly, SLITRK1 is primarily associated with the nervous system, including its involvement in the pathogenesis of Tourette's syndrome through cholinergic interneurons in mice, as well as its role in mediating neuropsychiatric behaviors related to chronic neuropathic pain (46, 47). We found that SLITRK1 plays a pivotal role in reducing the risk of obesity. In summary, the study identified three plasma proteins, FLT1, GAP43, and SLITRK1, as having a protective effect against obesity risk. Particularly, the findings revealed a previously unknown role for SLITRK1 in obesity prevention, expanding our understanding of the molecular mechanisms underlying this complex metabolic disorder and shedding light on potential targets for therapeutic interventions against obesity.

Our research has several limitations. First, the GWAS databases used primarily represent European demographics, which limits applicability to other ethnicities. Second,

we relied on a single database to evaluate the impacts of gut microbiota and metabolites on obesity without additional data sets for confirmation. Third, a lack of foundational experimental studies means our findings should be considered preliminary. Furthermore, SNP selection thresholds in MR such as for gut microbiota, metabolites, and plasma proteins, for which we used a looser $P < 1 \times 10^{-5}$, may affect the robustness of the results. We also did not apply multiple testing corrections, highlighting the need for further validation, and the absence of gene expression in obesity data complicates the assessment of the MR core hypothesis, suggesting that additional studies are needed.

We employed mediation MR analysis to explore the strong association between the latest GWAS findings on 473 gut microbiota and 1,400 plasma metabolites, and obesity. Our findings provide evidence for a causal relationship between seven specific bacterial species and obesity, which is mediated by metabolites. By analyzing genes associated with SNPs causally linked to obesity, gut microbiota, and metabolites, we found that FLT1, GAP43, and SLITRK1 proteins potentially reduce the risk of obesity. The findings enhance our understanding of obesity's complex mechanisms and identify new biomarkers and potential therapeutic targets. Future treatments may involve manipulating the gut microbiota and these plasma proteins to prevent or treat obesity.

## ACKNOWLEDGMENTS

This research did not receive any specific grant from funding agencies in the public, commercial, or not-for-profit sectors.

The research initiative and its execution were a joint effort between S.L. and X.L. X.L. and Q.W. undertook the statistical analysis and data interpretation. X.L. authored the manuscript, with substantial input and guidance from S.L., who oversaw the writing process. All contributors have provided their approval for the final submission of the paper.

## AUTHOR AFFILIATIONS

[1]Department of Nutrition and Food Hygiene, College of Public Health, Key Laboratory of Precision Nutrition and Health, Ministry of Education, Harbin Medical University, Heilongjiang, China
[2]Graduate School Capital Medical University, Beijing, China

## AUTHOR ORCIDs

Shan-peng Liu http://orcid.org/0009-0003-1643-0208

## AUTHOR CONTRIBUTIONS

Xiaomin Li, Data curation, Formal analysis, Methodology, Resources, Software, Writing – original draft | Qike Wu, Investigation, Methodology, Validation, Visualization | Shan-peng Liu, Conceptualization, Project administration, Resources, Supervision, Writing – review and editing

## ADDITIONAL FILES

The following material is available online.

### Supplemental Material

**Supplemental tables (Spectrum01892-24-S0001.xlsx).** Tables S1 to S7.

### Open Peer Review

**PEER REVIEW HISTORY (review-history.pdf).** An accounting of the reviewer comments and feedback.

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
