## [Reviewer comments · Microbiology Spectrum]

Microbiology Spectrum

Identification of 1,400 Metabolites as Mediators of Obesity in 473 Gut Microbiota Taxa: A Mediation Mendelian Randomization Study

Xiaomin Li, Qike Wu, and Shan-peng Liu

Corresponding Author(s): Shan-peng Liu, Capital Medical University

Review Timeline:

Submission Date:	July 29, 2024
Editorial Decision:	October 4, 2024
Revision Received:	October 15, 2024
Editorial Decision:	November 12, 2024
Revision Received:	November 22, 2024
Accepted:	December 17, 2024

Editor: Jinxin Liu

Reviewer(s): The reviewers have opted to remain anonymous.

Transaction Report:

DOI: <https://doi.org/10.1128/spectrum.01892-24>

Re: Spectrum01892-24 (**Identification of 1400 Metabolites as a Mediator of 473 Gut Microbiota-related Obesity: A Mediation Mendelian Randomization Study**)

Dear Dr. Shan-peng Liu:

Thank you for the privilege of reviewing your work. Below you will find my comments, instructions from the Spectrum editorial office, and the reviewer comments.

Revision Guidelines

Sincerely,
Jinxin Liu
Editor
Microbiology Spectrum

Reviewer #1 (Comments for the Author):

The innovation and value of the research are limited.

1. In abstract line 40, was the word "Fms" an abbreviation for something or wrong spelling?
2. Please correct the incorrect grammar and wording. eg. (Line 359) We finding.....

And the language of the full text needs to be improved.

3. In line 94, in terms of research design, what type of research is a "comprehensive study" or what's the definition of "comprehensive study"?
4. For names of bacteria, please use italics.
5. The study identified the gut microbiota, plasma metabolites and proteins associated with obesity, and suggested that the causal relationship between these seven specific microbiota and obesity is mediated by metabolites. However, the manuscript only explained their own relationship with obesity in the discussion and did not give a detailed description of how metabolites play the mediating role. Please make additional explanations in the manuscript.

This study employs bidirectional and two-sample Mendelian Randomization (MR) methods to explore the causal relationships between gut microbiota, metabolites, and obesity, identifying plasma proteins with protective effects, thereby suggesting potential targets for future treatment strategies. However, there are several key and technical issues in the text that require further clarification and improvement by the authors. Below are my main and minor comments.

Main Concerns

1. Comprehensiveness of the Dataset

- Is the obesity dataset used in the study sufficient in terms of sample size and SNP association? Have other obesity datasets been considered for validation?

2. Consideration of Confounding Factors

- What potential confounding factors have been considered in the study? What specific measures have been taken to mitigate these confounders?

3. Selection of Instrumental Variables

- Choosing 1×10^{-5} as the threshold for instrumental variables might not be stringent enough, potentially increasing the interference of confounders. Please explain the rationale for selecting this threshold.

4. Handling of Weak Instrumental Variables

- Has the article detailed how weak instrumental variables were selected and handled?

5. Criteria for Positive MR Analysis Results

- What are the criteria for determining positive results in MR analysis?

Consider adjusting the p-values in the results to reduce false positives and enhance the robustness of the analysis.

6. Relationship Between SNPs and Genes

- The article identifies SNPs related to obesity-associated gut microbiota/metabolites using MR analysis and then identifies related proteins using the vauils package. Given that the relationship between SNPs and nearby genes, as well as the specific effects of SNPs on genes, is unclear and not detailed in the text, if obesity-related SNPs are associated with certain proteins, this may conflict with the second and third MR assumptions (independence and exclusion restriction). Please provide a detailed explanation.

7. Results of Reverse MR Analysis

- Please explain the results of the reverse MR analysis. Were these results expected? What might these findings suggest?

Minor Concerns

1. Textual Accuracy

- Please correct the description of the three main assumptions of Mendelian Randomization in the text (line 141).

2. Format and Language

- The format of sentences and figures in the text needs further optimization, for example, changing "(Fig. 7A.)" to "(Fig. 7A)" and simplifying "single

nucleotide polymorphisms (SNPs)" to "SNPs" in line 302.

3. Clarity of Results Description

- In the results section 3.1, please clearly list the methods used along with the corresponding p-values and OR values.

Reviewer #1

Thank you so much for your positive and constructive comments, which have helped us to significantly improve the manuscript.

Point #1: In abstract line 40, was the word "Fms" an abbreviation for something or wrong spelling?

Response:

The term "Fms" in "Fms related receptor tyrosine kinase 1 (FLT1)" refers to its relation to the "Fms" family of receptors, which includes the macrophage colony-stimulating factor receptor. It is not a misspelling but rather denotes its association with this family of receptor tyrosine kinases.

Fms related receptor tyrosine kinase 1 (FLT1), also known as vascular endothelial growth factor receptor 1 (VEGFR-1), is an important receptor associated with angiogenesis and cell signal transduction (<https://www.ncbi.nlm.nih.gov/gene/2321>, https://www.genenames.org/data/gene-symbol-report/#!/hgnc_id/HGNC:3763)

This gene encodes a member of the vascular endothelial growth factor receptor (VEGFR) family. VEGFR family members are receptor tyrosine kinases (RTKs) which contain an extracellular ligand-binding region with seven immunoglobulin (Ig)-like domains, a transmembrane segment, and a tyrosine kinase (TK) domain within the cytoplasmic domain. This protein binds to VEGFR-A, VEGFR-B and placental growth factor and plays an important role in angiogenesis and vasculogenesis. Expression of this receptor is found in vascular endothelial cells, placental trophoblast cells and peripheral blood monocytes. Multiple transcript variants encoding different isoforms have been

found for this gene. Isoforms include a full-length transmembrane receptor isoform and shortened, soluble isoforms.

Point #2: Please correct the incorrect grammar and wording. eg. (Line 359)
We finding.....

And the language of the full text needs to be improved.

Response: We have addressed the grammatical errors and improved the language throughout the text. For example, we corrected the phrase "We finding unveiled the pivotal role of SLITRK1 in reducing the risk of obesity." to "We found that SLITRK1 plays a pivotal role in reducing the risk of obesity." We appreciate your suggestions, which have helped enhance the clarity and quality of our manuscript.

Point #3: In line 94, in terms of research design, what type of research is a "comprehensive study" or what's the definition of "comprehensive study"?

Response: The term "comprehensive study" is used to convey the diversity of our analytical methods for examining the interactions between gut microbiota, metabolites, and obesity, which includes two-sample, bidirectional, and mediated Mendelian randomization analyses, as well as enrichment analyses and PPI interaction networks. To eliminate ambiguity and enhance clarity, we will remove the term "comprehensive."

Point #4: For names of bacteria, please use italics.

Response: Thank you for your suggestion regarding the formatting of bacterial names. We appreciate your attention to detail, and we will ensure that all names of bacteria are presented in italics throughout the manuscript.

Point #5: The study identified the gut microbiota, plasma metabolites and proteins associated with obesity, and suggested that the causal relationship between these seven specific microbiota and obesity is mediated by metabolites. However, the manuscript only explained their own relationship with obesity in the discussion and did not give a detailed description of how metabolites play the mediating role. Please make additional explanations in the manuscript.

Response:

We appreciate your insight and recognize the need for a more comprehensive explanation in our manuscript. In the revised discussion, we will elaborate on the mechanisms through which these identified metabolites mediate the relationship between the specific gut microbiota and obesity.

“MR analyses reveal that metabolites such as oleoyl-linoleoyl-glycerol, phosphate to cysteine ratio, and alpha-ketoglutarate to kynurenine ratio mediate the relationship between gut microbiota and obesity. Gut bacteria can influence lipid metabolism and energy expenditure by producing molecules that affect fat storage (36). Additionally, adipocyte glutamine turnover is crucial for metabolic health, and the glutamine conjugate of C₆H₁₀O₂ may impact metabolic signaling pathways (37). Cysteine is emerging as a potential biomarker for obesity due to its role in antioxidant defense (38). Moreover, alpha-ketoglutarate supplementation promotes beige adipogenesis and reduces high-fat diet-induced obesity in middle-aged mice (39). Metabolites also interact with the gut-brain axis, affecting hunger and energy regulation (40).”

Reviewer #2

Thank you so much for your favorable comments and constructive feedback, which further enhanced our manuscript quality.

Main Concerns

Point #1: Comprehensiveness of the Dataset

Is the obesity dataset used in the study sufficient in terms of sample size and SNP association? Have other obesity datasets been considered for validation?

Response:

Thank you for your inquiry regarding the comprehensiveness of the obesity dataset used in our study. We would like to clarify that the obesity outcome data is sourced from the R10 version of the comprehensive Finnish database (<https://www.finngen.fi/>), which includes data from 412,055 participants. This large sample size enhances the statistical power of our findings and supports the robustness of the SNP associations we examined.

We believe that this dataset is sufficiently comprehensive for our analysis. However, we also acknowledge the importance of validation. While our study primarily focuses on the Finngen dataset, we are open to exploring the use of additional obesity datasets for validation purposes in future work. Therefore, we address this limitation in the Discussion section.

Point #2: Consideration of Confounding Factors

What potential confounding factors have been considered in the study? What specific measures have been taken to mitigate these confounders?

Response:

In our study, we acknowledged several potential confounding factors that could influence the relationships between gut microbiota, metabolites, and obesity. Here are the considerations and measures taken to address these confounders:

Potential Confounding Factors

1. **Genetic Variability:** Genetic differences among populations can affect gut microbiota composition and metabolic pathways.
2. **Environmental Factors:** Lifestyle factors such as diet, physical activity, and socioeconomic status can also play significant roles in obesity.
3. **Medications:** The use of antibiotics or other medications can alter gut microbiota, potentially confounding the results.
4. **Age and Sex:** These demographic factors can influence metabolism and microbiota composition.
5. **Existing Health Conditions:** Co-occurring health conditions may independently affect metabolism and obesity risk.

Mitigation Measures

1. **Mendelian Randomization (MR):** We used MR to exploit genetic variants as instrumental variables, which inherently control for confounding due to unobserved variables. This method minimizes the potential impact of both observed and unobserved confounding factors.
2. **Selection Criteria for SNPs:** We chose SNPs with a threshold of $P < 1 \times 10^{-5}$ for gut microbiota and metabolites and $P < 5 \times 10^{-8}$ for obesity outcomes, ensuring robust associations that are less likely to be influenced by confounders.
3. **Sensitivity Analyses:** Multiple analytical methods (e.g., MR-Egger, Weighted Mode, Weighted Median) were used to assess the robustness of our findings and detect possible pleiotropy or heterogeneity, which could suggest confounding effects.

4. Using publicly available GWAS databases, such as IEU Open GWAS and GWAS Catalog, allows researchers to search for all phenotype information related to specific SNPs in order to identify potential confounding factors. If any SNPs are found to be associated with known confounding factors, those SNPs will be excluded from further MR analysis. SNPs also were filtered using the Phenoscanner website (www.phenoscaner.medschl.cam.ac.uk).

5. Reverse Causation Analysis: To rule out reverse causality, we performed reverse MR analyses, ensuring that the identified associations were indeed causal and not a result of confounding due to existing obesity.

We discussed the limitations posed by the reliance on GWAS data predominantly representing European ancestries, highlighting the need for caution when generalizing findings to other populations. At the same time, we supplement the Methods section with relevant exclusion of confounders methods. By implementing these methods, we aimed to effectively mitigate the impacts of potential confounders and ensure that the relationships we observed were more likely to reflect true causal links between gut microbiota, metabolites, and obesity. Further research with diverse populations and additional datasets will enhance the robustness of our conclusions.

Point #3: Selection of Instrumental Variables

Choosing 1×10^{-5} as the threshold for instrumental variables might not be stringent enough, potentially increasing the interference of confounders. Please explain the rationale for selecting this threshold.

Response:

Thank you for your feedback regarding the selection of 1×10^{-5} as the threshold for instrumental variables. I would like to provide further clarification on our rationale for this choice.

Regarding obesity, we use a strict 1×10^{-8} threshold. However, we acknowledge that in some cases, using the "gold standard" threshold of 1×10^{-8} may not be practical, especially for certain non-disease complex human behavioral traits (gut microbiota, metabolites and plasma protein), where the genome-wide significant loci in GWAS results are relatively few. After the Clumping process, the remaining effective instrumental variables may be very limited, or even non-existent. This can lead to decreased statistical power and issues with weak instrumental variables, which can bias parameter estimates.

Thus, we selected 1×10^{-5} as the threshold to ensure a sufficient number of instrumental variables while minimizing the problems associated with weak instruments. This threshold allows us to extract more relevant SNPs, enhancing the statistical power of our analysis and ensuring that our results are more reliable. Additionally, we employed multiple methods in subsequent sensitivity analyses and robustness checks, including MR-Egger and MR-PRESSO, to further validate the robustness of our findings and to assess potential horizontal pleiotropy and heterogeneity.

We understand your concerns regarding the rigor of the threshold selection, and we will discuss this in more detail in the revised manuscript to convey our rationale more clearly. Thank you for your suggestions.

Point #4: Handling of Weak Instrumental Variables

Has the article detailed how weak instrumental variables were selected and handled?

Response: Thank you for your feedback on our handling of weak instrumental variables. In the methods section, we clarified that the f-statistic for each SNP was calculated to assess the strength of the instrumental variables. Specifically, SNPs with F values less than 10 were excluded to reduce weak

instrument bias. This approach is in line with established guidelines for ensuring the reliability of Mendelian randomization analyses.

Point #5: Criteria for Positive MR Analysis Results

What are the criteria for determining positive results in MR analysis? Consider adjusting the p-values in the results to reduce false positives and enhance the robustness of the analysis.

Response:

Thank you for your insightful question regarding the criteria for determining positive results in Mendelian Randomization (MR) analysis. In our study, we followed established guidelines for assessing positive MR results, which include the following criteria:

Statistical significance: We applied a stringent significance threshold of 1×10^{-5} for all MR associations. This threshold helps minimize the risk of false positives.

Consistency across methods: We employed multiple MR methods, including Inverse Variance Weighted, MR-Egger, Weighted Mode, and Weighted Median approaches. We viewed consistent results across these methods as supportive evidence for positive MR associations.

F-statistic: To assess the strength of our instrumental variables (SNPs), we calculated the F-statistic for each SNP. We excluded SNPs with F values less than 10 since weak instruments can lead to biased estimates. Strong instruments enhance the reliability of the causal inference.

Assessment of pleiotropy: We evaluated for horizontal pleiotropy using MR-Egger regression and MR-PRESSO tests. The absence of significant pleiotropic effects supports the validity of our MR assumptions and strengthens our positive findings.

Causal directionality: We conducted reverse MR analyses to verify the direction of causation, ensuring that the observed associations are not influenced by reverse causality.

Sensitivity analysis: We performed sensitivity analyses to assess the robustness of our results, exploring potential biases and ensuring that our findings remained stable under different analytical frameworks.

By adhering to these criteria, we aimed to ensure that our findings from the MR analyses are robust and reliable.

Point #6: Relationship Between SNPs and Genes

The article identifies SNPs related to obesity-associated gut microbiota/metabolites using MR analysis and then identifies related proteins using the vauils package. Given that the relationship between SNPs and nearby genes, as well as the specific effects of SNPs on genes, is unclear and not detailed in the text, if obesity-related SNPs are associated with certain proteins, this may conflict with the second and third MR assumptions (independence and exclusion restriction). Please provide a detailed explanation.

Response:

We utilized the vauils package to identify the genes most closely associated with obesity-related SNPs. The `find_nearest_gene` function in the vauils package allows for the rapid identification of the nearest genes. The vauils package requires information on the chromosome where the SNP is located, the position of the SNP on that chromosome, and SNP identifier. If a SNP appeared to correspond to more than one gene, the closest gene was selected, duplicates were removed and all genes associated with the SNP were merged. We apologize for the lack of detail in our article, which may have led to misunderstandings by reviewer 2. We clarify that we used the PPI interaction

network to further investigate the protein-level interaction relationships of SNP-related genes, rather than directly using the `vaults` package to identify related proteins. We have added more information to the methods section regarding the SNP-related genes.

The code is as follows:

1. Essential Library Imports

The code begins by loading required libraries:

```
library(vaults) # For function to find nearest genes
```

```
library(dplyr) # For data manipulation
```

2. Preparing the Data

The dataset containing SNP information is read into R from a CSV file and transformed into a data frame:

```
data <- read.csv('.csv', header = TRUE) # Reads CSV file into a data frame
```

```
data <- as.data.frame(data) # Ensures it's a data frame
```

```
data1 <- dplyr::select(data, 1, 2, 3) # Selects the first three columns  
(Chromosome number, Base position, SNP identifier )
```

3. Finding Nearest Genes

The core part of the code uses `find_nearest_gene` to identify genes located near the SNPs:

```
result <- find_nearest_gene(data1,
```

```
flanking = 50, # Specifies to search 50 kb upstream and downstream
```

```
build = "hg38", # Specifies the genomic build
```

```
collapse = TRUE, # If multiple genes are found, they will be collapsed into a  
single entry
```

```
snp = "SNP", # Column name for SNP identifiers
```

```
chr = "CHR", # Column name for chromosome
```

```
bp = "POS") # Column name for base pair positions
```

Point #7: Results of Reverse MR Analysis

Please explain the results of the reverse MR analysis. Were these results

expected? What might these findings suggest?

Response:

Through reverse MR analysis, we examine whether obesity influences specific gut microbiota and metabolites. The verification of unidirectionality in causal relationships indicates that gut microbiota and metabolites have a direct influence on obesity, whereas the effect of obesity on these microbiota and metabolites is comparatively limited. By testing reverse causation and confirming unidirectionality, we can more accurately assess the roles of various factors in the pathogenesis of obesity. If reverse MR causality exists, a causal relationship between gut microbiota or metabolites on obesity cannot be proven.

Minor Concerns

Point #1: Textual Accuracy

Please correct the description of the three main assumptions of Mendelian Randomization in the text (line 141).

Response:

Thank you for your valuable feedback. We have made the requested adjustments to the description of the three main assumptions of Mendelian Randomization in the text. The revised assumptions are as follows:

(1) genetic variant must reliably associate with the risk factor; (2) genetic variant must not associate with any known or unknown confounders; (3) genetic variant must influence the outcome only through the risk factor and not through any direct causal pathway.

Point #2: Format and Language

The format of sentences and figures in the text needs further optimization, for example, changing "(Fig. 7A.)" to "(Fig. 7A)" and simplifying "single nucleotide polymorphisms (SNPs)" to "SNPs" in line 302.

Response:

Thank you for your constructive feedback regarding the formatting of sentences and figures. We have made the necessary adjustments throughout the manuscript, including changing "(Fig. 7A.)" to "(Fig. 7A)" and simplifying "single nucleotide polymorphisms (SNPs)" to "SNPs" in line 302. Additionally, we have reviewed the entire text for similar formatting errors and corrected them accordingly.

Point #3: Clarity of Results Description

In the results section 3.1, please clearly list the methods used along with the corresponding p-values and OR values.

Response:

We have revised this section to clearly outline the methods used, along with the corresponding p-values and odds ratios (OR values). Additionally, we have reviewed the entire text for similar formatting errors and corrected them.

Re: Spectrum01892-24R1 (**Identification of 1400 Metabolites as a Mediator of 473 Gut Microbiota-related Obesity: A Mediation Mendelian Randomization Study**)

Dear Dr. Shan-peng Liu:

Thank you for the privilege of reviewing your work. Below you will find my comments, instructions from the Spectrum editorial office, and the reviewer comments.

Revision Guidelines

Sincerely,
Jinxin Liu
Editor
Microbiology Spectrum

Reviewer #1 (Comments for the Author):

The author's reply addressed my concerns.

Reviewer #2 (Comments for the Author):

Thank you for your detailed responses to the reviewer comments and for making corresponding revisions to the manuscript. However, there are still some aspects that require further modification to enhance the rigor and completeness of the paper. Specifically, the following points need additional improvement:

1. In the article, the SNP selection thresholds differ between forward and reverse MR analyses (2.3 Instrumental variable selection). Is this discrepancy in selection criteria reasonable? Could this be the reason for the negative results in the reverse MR analysis?
2. The author specifies in the response to Point #5 that the statistical significance threshold for positive results is set at 1×10^{-5} , but in the results section 3.1, the significance threshold is 0.05. Please explain.
3. In the two MR analyses (gut microbiota and obesity; metabolites and obesity), were the p-values of the study results adjusted? Was multiple testing correction performed in the study?
4. The author's response to Point #6 (Relationship Between SNPs and Genes) is not sufficiently clear, and it seems the author may have misunderstood my question. In the article, two MR analyses were first conducted to identify the causal relationship between various gut microbiota and metabolites with obesity, leading to the identification of relevant SNPs associated with gut microbiota and metabolites. These SNPs, as instrumental variables for gut microbiota and metabolites, need to satisfy the three core assumptions of MR analysis, namely the association assumption, the independence assumption, and the exclusion restriction assumption. The author subsequently derived the genes associated with these SNPs based on their positional information. If the SNPs are related to these genes, does this violate the three core assumptions of MR?

Dear Authors,

Thank you for your detailed responses to the reviewer comments and for making corresponding revisions to the manuscript. However, there are still some aspects that require further modification to enhance the rigor and completeness of the paper. Specifically, the following points need additional improvement:

1. In the article, the SNP selection thresholds differ between forward and reverse MR analyses (2.3 Instrumental variable selection). Is this discrepancy in selection criteria reasonable? Could this be the reason for the negative results in the reverse MR analysis?
2. The author specifies in the response to Point #5 that the statistical significance threshold for positive results is set at 1×10^{-5} , but in the results section 3.1, the significance threshold is 0.05. Please explain.
3. In the two MR analyses (gut microbiota and obesity; metabolites and obesity), were the p-values of the study results adjusted? Was multiple testing correction performed in the study?
4. The author's response to Point #6 (Relationship Between SNPs and Genes) is not sufficiently clear, and it seems the author may have misunderstood my question. In the article, two MR analyses were first conducted to identify the causal relationship between various gut microbiota and metabolites with obesity, leading to the identification of relevant SNPs associated with gut microbiota and metabolites. These SNPs, as instrumental variables for gut microbiota and metabolites, need to satisfy the three core assumptions of MR analysis, namely the association assumption, the independence assumption, and the exclusion restriction assumption. The author subsequently derived the genes associated with these SNPs based on their positional information. If the SNPs are related to these genes, does this violate the three core assumptions of MR?

Reviewer #2

Thank you so much for your positive and constructive comments, which have helped us to significantly improve the manuscript.

Point #1: In the article, the SNP selection thresholds differ between forward and reverse MR analyses (2.3 Instrumental variable selection). Is this discrepancy in selection criteria reasonable? Could this be the reason for the negative results in the reverse MR analysis?

Response:

Thank you for your valuable question regarding the discrepancy in SNP selection thresholds between the forward and reverse MR analyses. In our response to the previous Question 3, we mentioned that for obesity, we employ a stringent threshold of 1×10^{-8} . However, we recognize that applying this "gold standard" threshold may not be practical for certain complex traits, such as gut microbiota, metabolites, and plasma proteins, which often have fewer genome-wide significant loci in GWAS results.

To mitigate issues associated with limited effective instrumental variables after the clumping process, we adopted a threshold of $P < 1 \times 10^{-5}$. This choice was made to ensure we retained a sufficient number of SNPs while minimizing the potential pitfalls of weak instruments, which could bias our estimates and reduce statistical power. The $P < 1 \times 10^{-5}$ threshold is supported by multiple studies in the literature and allows us to extract more relevant SNPs, thereby enhancing the reliability of our results (1, 2).

We also conducted multiple sensitivity analyses and robustness checks, including MR-Egger and MR-PRESSO, to further validate our findings and assess potential horizontal pleiotropy and heterogeneity in our analyses.

We appreciate your feedback and we have stated the limitations in the discussion section to improve the rigor of the paper.

Point #2: The author specifies in the response to Point #5 that the statistical significance threshold for positive results is set at 1×10^{-5} , but in the results section 3.1, the significance threshold is 0.05. Please explain.

Response:

Thank you for your careful review and for pointing out the discrepancy in our manuscript. We acknowledge that there was an error in our previous response regarding the significance threshold for positive results.

To clarify, the significance threshold of 0.05 mentioned in section 3.1 of the results is indeed correct. This threshold is used for determining statistical significance in the context of the analyses presented. The $P < 1 \times 10^{-5}$ threshold discussed in our response was intended for the selection of SNPs as instrumental variables, which is a separate consideration from the overall statistical significance in the results.

Point #3: In the two MR analyses (gut microbiota and obesity; metabolites and obesity), were the p-values of the study results adjusted? Was multiple testing correction performed in the study?

Response:

Thank you for your important question regarding the adjustment of p-values and the application of multiple testing correction in our MR analyses.

We acknowledge that we did not perform multiple testing correction in our study. This decision was based on the exploratory nature of our research, which is aimed at identifying potential associations rather than confirming

specific hypotheses. In addition, there is existing literature supporting the rationale for not applying strict multiple testing corrections in exploratory studies, as it allows for the discovery of potential positive findings that can be investigated further in subsequent research (3-5).

We understand the importance of addressing this potential limitation and have elaborated on this point in the limitations section of the discussion.

Point #4: The author's response to Point #6 (Relationship Between SNPs and Genes) is not sufficiently clear, and it seems the author may have misunderstood my question. In the article, two MR analyses were first conducted to identify the causal relationship between various gut microbiota and metabolites with obesity, leading to the identification of relevant SNPs associated with gut microbiota and metabolites. These SNPs, as instrumental variables for gut microbiota and metabolites, need to satisfy the three core assumptions of MR analysis, namely the association assumption, the independence assumption, and the exclusion restriction assumption. The author subsequently derived the genes associated with these SNPs based on their positional information. If the SNPs are related to these genes, does this violate the three core assumptions of MR?

Response:

We appreciate your logical reasoning and inference concerning the relationship between the identified SNPs and the associated genes in the context of Mendelian Randomization.

In our analyses, we conducted both bidirectional MR and two-sample MR approaches, which allowed us to identify 548 SNPs that are causally associated with various gut microbiota/metabolites in relation to obesity.

Utilizing bidirectional Mendelian Randomization (MR) and two-sample MR approaches, we identified 548 SNPs associated with gut microbiota and

metabolites that are causally linked to obesity and inferred 381 genes based on the locations of these SNPs. The results of our Mendelian analysis indicate that the 548 SNPs are not directly related to the outcome of obesity, meaning that the 381 genes do not exhibit differential expression between obese and non-obese populations. However, we currently do not have expression data to investigate these 381 genes in the context of obesity versus non-obesity. If validation shows that these 381 genes do not have differential expression between obese and non-obese individuals, it means that the verification results could support the independence assumption and the exclusion restriction assumption of MR. Consequently, these findings require further validation. We have acknowledged this limitation in the Discussion section of the manuscript.

In summary, we should include the following limitations in the discussion section: “Furthermore, SNP selection thresholds in MR such as for gut microbiota, metabolites, and plasma proteins, for which we used a looser $P < 1 \times 10^{-5}$, may affect the robustness of the results. We also did not apply multiple testing correction, highlighting the need for further validation, and the absence of gene expression in obesity data complicates the assessment of the MR core hypothesis, suggesting that additional studies are needed.”

1. Li P, Wang H, Guo L, Gou X, Chen G, Lin D, et al. Association between gut microbiota and preeclampsia-eclampsia: a two-sample Mendelian randomization study. *BMC Med.* 2022;20(1):443.
2. Xiao G, He Q, Liu L, Zhang T, Zhou M, Li X, et al. Causality of genetically determined metabolites on anxiety disorders: a two-sample Mendelian randomization study. *J Transl Med.* 2022;20(1):475.
3. Zhang Y, Li D, Zhu Z, Chen S, Lu M, Cao P, et al. Evaluating the impact of metformin targets on the risk of osteoarthritis: a mendelian randomization study. *Osteoarthritis Cartilage.* 2022;30(11):1506-14.
4. Yuan S, Titova OE, Zhang K, Gou W, Schillemans T, Natarajan P, et al. Plasma protein and venous thromboembolism: prospective cohort and mendelian randomisation analyses. *Br J Haematol.* 2023;201(4):783-92.
5. Xiang Y, Zhang C, Wang J, Cheng Y, Wang L, Tong Y, et al. Identification of host gene-microbiome

associations in colorectal cancer patients using mendelian randomization. *J Transl Med.* 2023;21(1):535.

Re: Spectrum01892-24R2 (**Identification of 1400 Metabolites as a Mediator of 473 Gut Microbiota-related Obesity: A Mediation Mendelian Randomization Study**)

Dear Dr. Shan-peng Liu:

Your manuscript has been accepted, and I am forwarding it to the ASM production staff for publication. Your paper will first be checked to make sure all elements meet the technical requirements. ASM staff will contact you if anything needs to be revised before copyediting and production can begin. Otherwise, you will be notified when your proofs are ready to be viewed.

Sincerely,
Jinxin Liu
Editor
Microbiology Spectrum